# A motogenic GABAergic system of mononuclear phagocytes facilitates dissemination of coccidian parasites

Amol K Bhandage[1]*, Gabriela C Olivera[1], Sachie Kanatani[1], Elizabeth Thompson[2], Karin Loré[2], Manuel Varas-Godoy[3], Antonio Barragan[1]*

[1]Department of Molecular Biosciences, The Wenner-Gren Institute, Stockholm University, Stockholm, Sweden; [2]Department of Medicine Solna, Karolinska Institutet, Stockholm, Sweden; [3]Cancer Cell Biology Laboratory, Center for Cell Biology and Biomedicine (CEBICEM), Faculty of Medicine and Science, Universidad San Sebastian, Santiago, Chile

**Abstract** Gamma-aminobutyric acid (GABA) serves diverse biological functions in prokaryotes and eukaryotes, including neurotransmission in vertebrates. Yet, the role of GABA in the immune system has remained elusive. Here, a comprehensive characterization of human and murine myeloid mononuclear phagocytes revealed the presence of a conserved and tightly regulated GABAergic machinery with expression of GABA metabolic enzymes and transporters, GABA-A receptors and regulators, and voltage-dependent calcium channels. Infection challenge with the common coccidian parasites *Toxoplasma gondii* and *Neospora caninum* activated GABAergic signaling in phagocytes. Using gene silencing and pharmacological modulators *in vitro* and *in vivo* in mice, we identify the functional determinants of GABAergic signaling in parasitized phagocytes and demonstrate a link to calcium responses and migratory activation. The findings reveal a regulatory role for a GABAergic signaling machinery in the host-pathogen interplay between phagocytes and invasive coccidian parasites. The co-option of GABA underlies colonization of the host by a *Trojan horse* mechanism.

*For correspondence:
amol.bhandage@su.se (AKB);
antonio.barragan@su.se (AB)

Competing interests: The authors declare that no competing interests exist.

## Introduction

Gamma-aminobutyric acid (GABA), first identified as a plant and microbe metabolite, is a principal neurotransmitter in the central nervous system (CNS) of vertebrates (*Roth et al., 2003*). Moreover, recent findings implicate GABAergic signaling in the disease environment of cancer and other inflammatory conditions in humans (*Neman et al., 2014*; *Bhat et al., 2010*; *Takehara et al., 2007*; *Li et al., 2017*). Neurons and other GABAergic cells synthesize GABA via glutamate decarboxylases (GAD65/67) (*Soghomonian and Martin, 1998*). GABA is shuttled in and out of cells via GABA transporters (GATs) (*Höglund et al., 2005*) and acts via activation of GABA-A receptors (GABA-A Rs) (*Olsen and Sieghart, 2008*) and GABA-B Rs (*Bettler et al., 2004*). The GABA-A Rs are pentameric ionotropic chloride channels, normally comprised of three types of subunits: 2 αs, 2 βs and a third type of subunit. Nineteen different mammalian GABA-A R subunits (α1–6, β1–3, γ1–3, δ, ε, π, θ and ρ1–3) can combine to form numerous variants of functional heteromeric receptors in neuronal cells. The strength and polarity of GABA signaling is regulated by cation-chloride cotransporters (CCCs) (*Kaila et al., 2014*). GABA-A R activation by GABA can elicit opening of voltage-dependent calcium (Ca$^{2+}$) channels (VDCCs) with subsequent Ca$^{2+}$ influx into the neuronal cell (*Bortone and Polleux, 2009*).

Owing to host-pathogen coevolution with reciprocal selection, studies of host-pathogen interactions provide a powerful tool to gain insight into biological processes. The obligate intracellular

protozoan *Toxoplasma gondii* actively invades nucleated cells (*Sibley, 2004*) and has a broad range of hosts among warm-blooded vertebrates. One third of the global human population is estimated to be chronically infected by *T. gondii* (*Pappas et al., 2009*) and severe manifestations may occur upon immunosuppression and during pregnancy (*Montoya and Liesenfeld, 2004*). Similarly, the related coccidian *Neospora caninum* represents a pathogen of major importance in veterinary medicine (*Dubey et al., 2007*). Upon ingestion and after crossing the intestinal epithelium, the tachyzoite stages of these coccidian parasites rapidly disseminate in their intermediate hosts, ultimately establishing latent infection in the CNS (*Montoya and Liesenfeld, 2004*; *Dubey et al., 2007*).

The mononuclear phagocyte system comprises dendritic cells (DCs), monocytes, macrophages and brain microglia, which mediate multiple immunological functions and are crucial to counteract microbial infection (*Guilliams et al., 2014*). Early on during infection, tissue-invasive coccidian tachyzoites encounter DCs and other phagocytes, which play a determinant role in mounting a robust host-protective immune response (*Liu et al., 2006*; *Mashayekhi et al., 2011*). Paradoxically, *T. gondii* and *N. caninum* exploit the inherent migratory ability of DCs and monocytes for dissemination via a *Trojan horse* mechanism (*Sangaré et al., 2019*; *Lambert et al., 2006*; *Lambert et al., 2009*; *Courret et al., 2006*; *Collantes-Fernandez et al., 2012*). Within minutes of active invasion by *T. gondii*, DCs adopt a hypermigratory phenotype that mediates rapid systemic dissemination in mice (*Kanatani et al., 2017*; *Weidner and Barragan, 2014*). GABAergic inhibition of DCs hampers dissemination of *T. gondii* (*Kanatani et al., 2017*; *Fuks et al., 2012*), however the precise mechanisms of action have remained uncharacterized.

Some components of GABAergic signaling have been detected in DCs, monocytes, macrophages, T cells and B cells (*Bhat et al., 2010*; *Fuks et al., 2012*; *Alam et al., 2006*; *Wheeler et al., 2011*; *Bhandage et al., 2014*). Yet, the precise functions of GABA in immune cells have remained elusive. Here, we have performed a systematic analysis of the GABAergic system of various types of human and murine DCs and monocytes and report a conserved GABAergic machinery implicated in migratory responses. Further, we show that coccidian parasites hijack GABAergic signaling in parasitized phagocytes to promote infection-related dissemination.

## Results

### Human and murine mononuclear phagocytes exhibit hypermotility and secrete GABA upon challenge with *T. gondii* and *N. caninum*

To address the impact of coccidian infection on mononuclear phagocyte motility, primary cells from human donors and mice were challenged with freshly egressed tachyzoites of *T. gondii* and *N. caninum*. Upon invasion by tachyzoites (*Figure 1A*), a rapid increase of migrated distances (*Figure 1B*) and elevated velocities (*Figure 1C*) were recorded in infected murine bone marrow-derived DCs (mBMDCs), compared with unchallenged cells. Similarly, human monocytes, monocyte-derived DCs (hMoDCs) and primary myeloid DCs (hMDCs) freshly isolated from blood of human donors consistently exhibited hypermotility upon challenge (*Figure 1D,E,F*, *Figure 1—figure supplement 1A,B, C*). Further, phagocytes expressed transcripts of enzymes for GABA synthesis and catabolism (*Figure 1G*), which were up- and downregulated, respectively, upon challenge with *T. gondii* (*Figure 1H*). Consistent with this, elevations of GABA concentrations in the supernatants were detected shortly after challenge and increased over time (*Figure 1I*). Further, challenge of mononuclear phagocytes with separate strains of the two coccidia resulted in elevated concentrations of GABA in cell supernatants (*Figure 1J*). Together, this indicated a putative link between GABA and migratory activation, motivating a further analysis of GABAergic signaling in mononuclear phagocytes.

### A conserved repertoire of GABA-A R subunits in human and murine phagocytes

To address GABAergic signaling in phagocytes, human and murine cells were screened for expression of the 19 known GABA-A R subunits (*Supplementary file 1*), which in neuronal cells combine to form multiple pentameric receptor variants (*Olsen and Sieghart, 2008*). Interestingly, mBMDCs consistently expressed mRNAs for 10 out of the 19 GABA-A R subunits (α3, α4, α5, β2, β3, γ1, γ2, δ, ρ1,

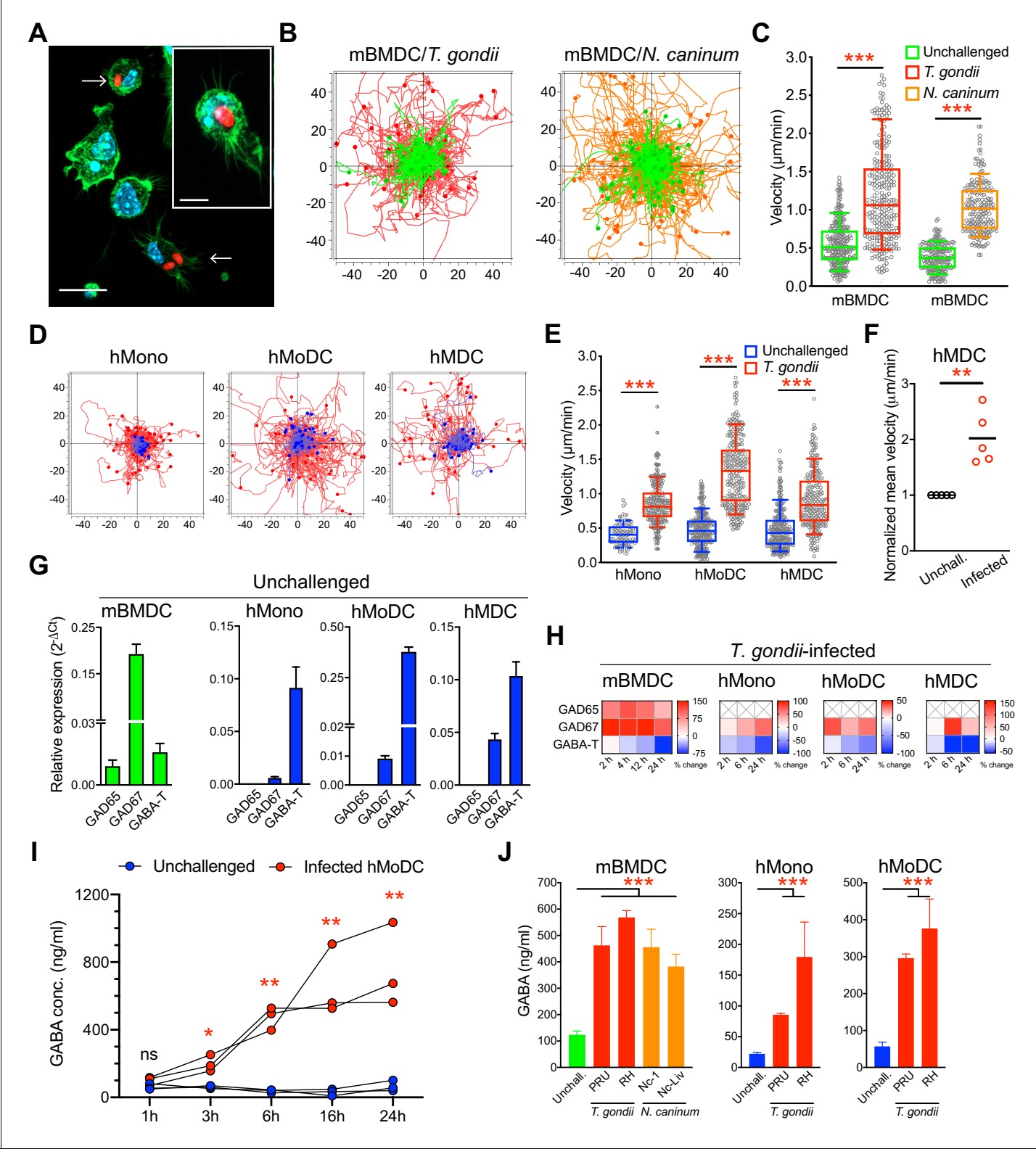

**Figure 1.** Migratory activation and GABAergic responses of mononuclear phagocytes challenged with *T. gondii* and *N. caninum*. (**A**) Representative micrographs of mBMDCs infected by *T. gondii* tachyzoites (PRU-RFP, arrowheads) and by-stander non-infected BMDCs, stained for F-actin (Alexa Fluor 488 Phalloidin) and nuclei (DAPI). Scale bars, 20 μm, inset image 10 μm. (**B**) Representative motility plots of unchallenged mBMDCs (green) and *T. gondii*- or *N. caninum*-infected mBMDCs (red and orange, respectively). X- and y-axes show distances in μm. (**C**) Box-and-whisker dot plots show, for

*Figure 1 continued*

each condition in (B), median velocities (µm/min) of cells shown in motility plots (n = 3–6 independent experiments). (D) Motility plots of hMonocytes, hMoDCs and hMDCs, respectively, unchallenged (blue) or infected with *T. gondii* (PRU-RFP, red). (E) Quantifications of velocities (µm/min) in (D) are shown as box-and-whisker dot plots (n = 3–6 independent experiments). (F) Normalized mean velocities of unchallenged and *T. gondii*-infected hMDCs from human donors (n = 5 independent donors, 50–60 cells per donor). (G) Relative mRNA expression ($2^{-\Delta Ct}$) of GABA synthesis enzymes (GAD65 and GAD67) and catabolic enzyme (GABA-T) in unchallenged mBMDCs, hMoDCs, hMDCs and hMonocytes, respectively, determined by real-time qPCR. (H) Corresponding heat maps for the indicated cell types show (%) transcriptional changes upon challenge with *T. gondii* relative to unchallenged cells at indicated time-points, as detailed in Materials and methods. (X) indicates no amplification (n = 3–6 independent experiments). (I) GABA (ng/ml) secreted in supernatants of hMoDCs challenged with *T. gondii* (ME49/PTG) at indicated time-points (n = 3 human donors). (J) GABA (ng/ml) secreted in supernatants of mBMDCs, hMonocytes and hMoDCs, respectively, challenged with different strains of *T. gondii* (PRU and RH) or *N. caninum* (Nc-1 and Nc-Liverpool) was quantified by ELISA (n = 6–8 independent experiments). Bar graphs show mean + SEM. Statistical significance was tested by Mann-Whitney test for (C, E, F), ordinary one-way ANOVA with Dunnett's multiple comparison test for (J) and paired t-test for (I), *p<0.05, **p<0.01, ***p<0.001, ns p≥0.05.

The online version of this article includes the following figure supplement(s) for figure 1:

**Figure supplement 1.** hMDCs exhibit a hypermotile phenotype upon *T. gondii* challenge.

---

ρ2) (*Figure 2A*, *Supplementary file 2*) and expression of GABA-A R subunits was conserved in a murine DC line (JAWSII) (*Figure 2—figure supplement 1A*). Further, hMoDCs from healthy donors consistently expressed the α4, ρ1, ρ2 and ρ3 subunits, while expression of the α3, β1 and β3 subunits was only detected in some donors (*Figure 2B*, *Supplementary file 2*). Finally, freshly isolated hMDCs from human donors consistently expressed β2, θ, ρ1 and ρ2 subunits of GABA-A Rs (*Supplementary file 2*). Interestingly, challenge with *T. gondii* modulated the subunit mRNA expression in murine and human DCs (*Figure 2C,D*, *Figure 2—figure supplement 1B*). Of note, transcriptional expression of the α6, β2 and θ subunits, undetectable in unchallenged hMoDCs, was consistently observed in challenged hMoDCs (*Figure 2B,D*). Immunocytochemical analyses with available antibodies yielded signal consistent with expression of α3, α5, β3 and ρ1 subunits (*Figure 2E*). We conclude that murine and human phagocytes constitutively transcribed one or more α subunits, one or more β subunits and one or more additional subunits, including ρ GABA-A R subunits (*Supplementary file 2*). Additionally, the modulated subunit expression upon challenge with *T. gondii* motivated an assessment of GABA-A R function upon infection.

## Selective pharmacological inhibition of GABA-A Rs abrogates *T. gondii-/N. caninum*-induced hypermotility in human and murine phagocytes

To functionally assess the putative implication of GABA-A Rs in parasite-induced migratory activation of DCs, motility assays were performed in the presence of general and subunit-selective GABA-A R antagonists and modulators. A broad range GABA-A R open-channel blocker (picrotoxin) and α, β and ρ subunit selective inhibitors (L655 708, SCS and TPMPA, respectively), efficiently inhibited *T. gondii-/N. caninum*-induced hypermotility in mBMDCs (*Figure 3A,C*, *Figure 3—figure supplement 1A,B*), hMoDCs (*Figure 3B,D*) and, freshly isolated hMDCs and monocytes (*Figure 1—figure supplement 1D,E*, *Figure 3—figure supplement 1C*), with a non-significant impact on baseline cell motility. Further, we took advantage of the finding that blockade of GABA transporters (SNAP) inhibited hypermotility in parasitized mBMDCs (*Figure 3E,F*) with reduced GABA concentrations in the supernatants (Figure 5F) to attempt reconstitution of hypermotility by allosteric modulators of GABA-A Rs. Importantly, hypermotility was rescued by an allosteric modulator of β2/β3-containing GABA-A Rs (etomidate), but not by an allosteric modulator of δ subunit-containing GABA-A Rs (allopregnanolone) (*Figure 3E,F*). Because mBMDCs transcriptionally express β2, β3 and δ subunits (*Figure 2*), the data jointly indicated an implication of β subunits in hypermotility and also differential effects or implication by specific subunits. Together, this indicated that GABA-A Rs are implicated in the motogenic activation of phagocytes upon coccidian challenge and, specifically raised the question of participation of α, β and ρ GABA-A Rs subunits in hypermotility of phagocytes.

## Gene silencing of GABA-A R subunits inhibits DC hypermotility

To determine which GABA-A R subunits were implicated in hypermotility, we designed a gene silencing approach in primary DCs. First, based on subunit expression data (*Figure 2*) and on the

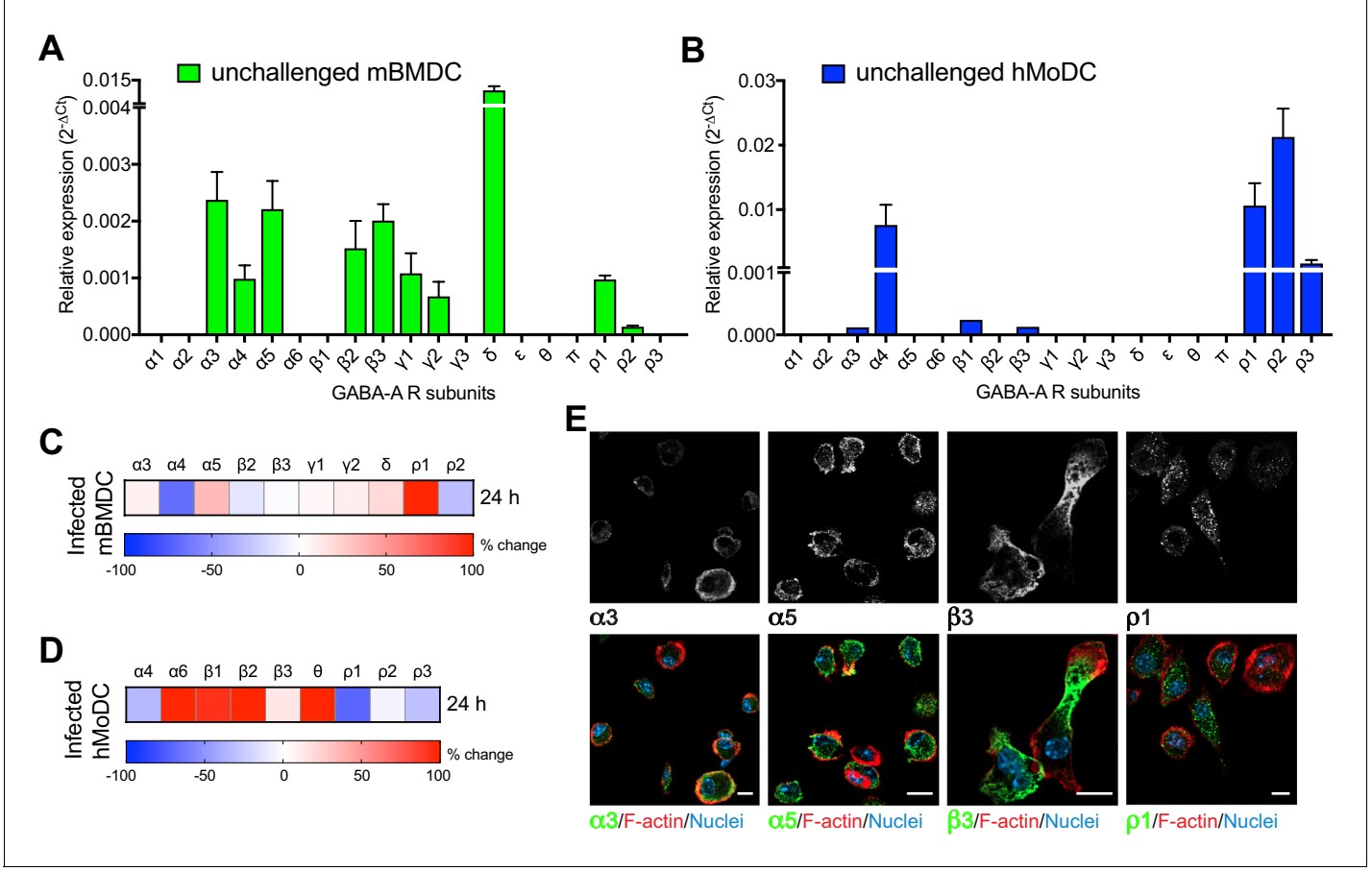

**Figure 2.** Modulated expression of GABA-A R subunits in DCs upon *T. gondii* infection. (**A, B**) Relative mRNA expression (Mean + SEM, $2^{-\Delta Ct}$) of GABA-A R subunits in unchallenged (**A**) mBMDCs (n = 7 independent experiments) and (**B**) hMoDCs (n = 3 independent experiments). (**C, D**) Heat maps show (%) transcriptional changes in the expression of GABA-A R subunits upon challenge of (**C**) mBMDCs and (**D**) hMoDCs with *T. gondii* (PRU-RFP) relative to unchallenged cells at indicated time-point (n = 3–4 independent experiments). (**E**) Representative micrographs of unchallenged mBMDCs stained with antibodies against GABA-A R α3, α5, β3 and ρ1 subunit (and Alexa Flour 488-conjugated secondary antibodies), respectively, and Alexa Fluor 647 Phalloidin (F-actin) and DAPI (nuclei). Scale bars, 10 μm, for β3 5 μm (n = 3 independent experiments).

The online version of this article includes the following figure supplement(s) for figure 2:

**Figure supplement 1.** Expression of GABA-A receptor subunits in JAWSII is modulated upon *T. gondii* infection.

impact of selective pharmacological inhibitors on migration (*Figure 3*), we designed probes (*Supplementary file 3*) to knock-down the mRNA expression of α3, β3 and ρ1 subunits in mBMDCs and the α4 and ρ2 subunits in hMoDCs, respectively. Second, the shRNA probes were validated in neuronal cell lines known to express GABA-A Rs (*Figure 4—figure supplement 1A–E*) and significantly silenced target transcription in mBMDCs (*Figure 4A*) and hMoDCs (*Figure 4B*). Finally, the impact of gene silencing on hypermotility was assessed. The hypermotility phenotypes remained unaffected in mock-transduced and control shRNA-treated conditions (*Ólafsson et al., 2019*; *Figure 4C,D,E,F*). In contrast, hypermotility was abolished in shβ3- and shρ1-treated mBMDCs (*Figure 4C,D,G*), demonstrating the implication of β3 and ρ1 subunits. In shα3-treated cells (with two separate constructs), hypermotility was significantly reduced, albeit not abolished, indicating a contribution by the α3 subunit. In hMoDCs, hypermotility was significantly reduced in shα4-treated cells, but not in shρ2-treated cells (*Figure 4E,F,G*), indicating primarily a dependence on the α4 subunit for hypermotility. Jointly with pharmacological inhibition, these data demonstrate a critical dependency of DC hypermotility on the β3 and ρ1 subunits for mBMDCs and on the α4 subunit for hMoDCs.

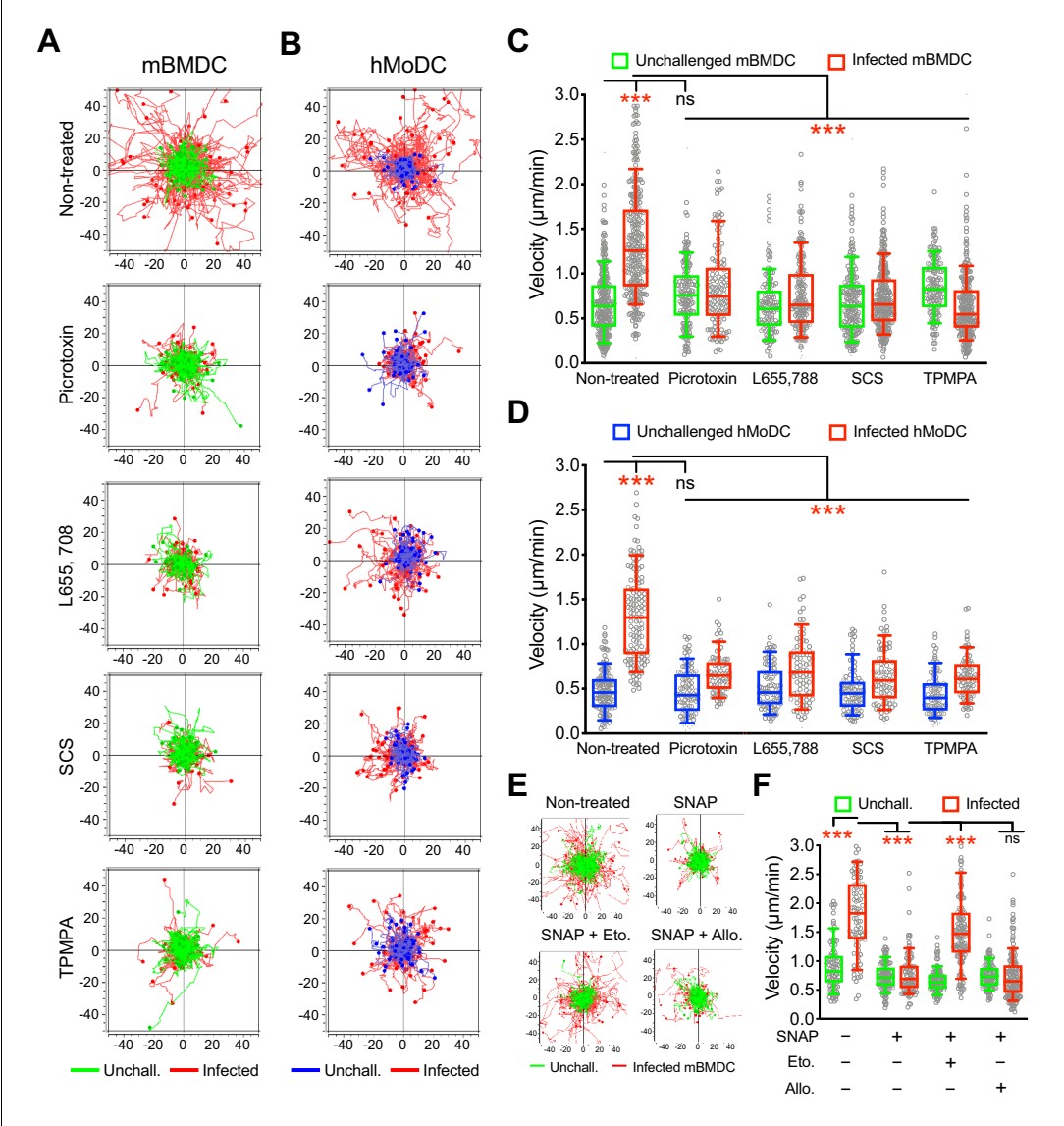

**Figure 3.** Impact of pharmacological modulation of GABA-A Rs on *T. gondii*-induced hypermotility of DCs. (A, B) Representative motility plots of unchallenged mBMDCs and *T. gondii* (PRU-RFP)-infected (A) mBMDCs and (B) hMoDCs treated with GABA-A R inhibitor picrotoxin (open channel blocker) or subunit specific inhibitors L655,708 (α-specific), SCS (β-specific) and TPMPA (ρ-specific) at concentrations stated in Materials and methods. X- and y-axes indicate distances in μm. (C, D) Box-and-whisker dot plots show, for each condition, median velocities (μm/min) as indicated in (A) and (B) (n = 3–6 independent experiments). (E) Representative motility plots and (F) velocities of unchallenged mBMDCs and *T. gondii*-infected mBMDCs treated with SNAP (GAT inhibitor) in presence of etomidate and allopregnanolone (allosteric modulators of β and δ subunit-containing GABA-A Rs, respectively). (n = 3–4 independent experiments). Statistical significance was tested by ordinary one-way ANOVA with Dunnett's multiple comparison test for (C, D, F), ***$p < 0.001$, ns $p \geq 0.05$.

The online version of this article includes the following figure supplement(s) for figure 3:

**Figure supplement 1.** Effect of pharmacological inhibitors of GABA-A Rs, NKCC1 and VDCCs on hypermotility.

## Abolished DC hypermotility by inhibition of GABA synthesis (GAD67) and secretion is rescued by GABA-A R agonism

The constitutive elements of a GABA synthesis and transport in human DCs have remained uncharacterized (*Fuks et al., 2012*). In addition to GABA synthesis enzymes (*Figure 1G*), hMoDCs and mBMDCs expressed GABA transporters (*Figure 5A, C*), whose expression was modulated by challenge with *T. gondii* (*Figure 5B, D*). Pharmacological inhibition of GABA synthesis (SC) (*Figure 5E, H*) or transportation (SNAP) (*Figure 5E, G*) inhibited hypermotility and reduced GABA

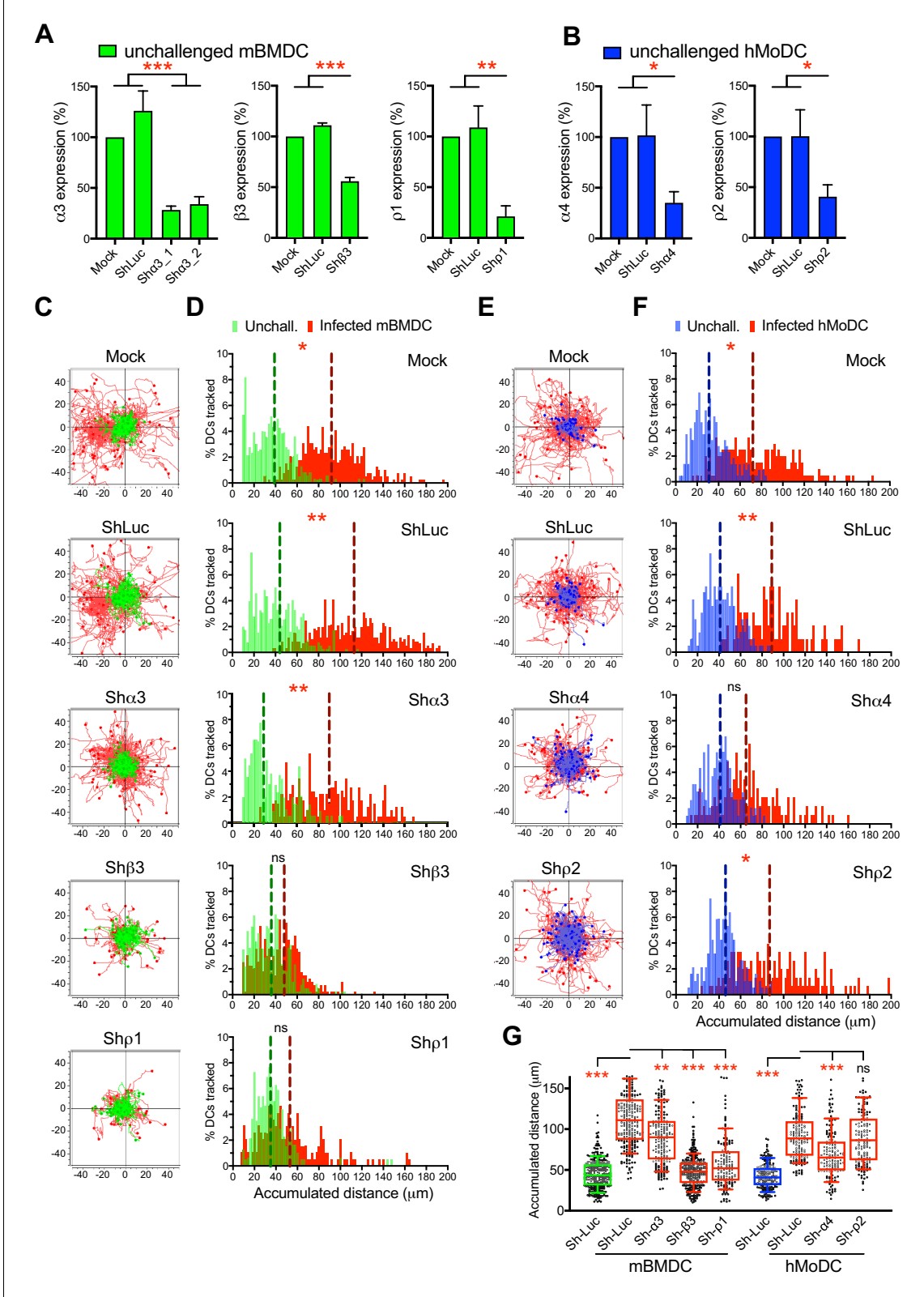

**Figure 4.** Targeted gene silencing of GABA-A R subunits impacts DC hypermotility. (A, B) The mRNA expression in unchallenged (A) mBMDCs and (B) hMoDCs, treated with shRNA for control Luc and GABA-A R subunits, related to mock-treated cells (Mean + SEM, %). For shα3, two separate constructs were used (n = 3–5 independent experiments for mBMDCs and 6–7 for hMoDCs). (C, E) Representative motility plots of (C) mBMDCs and (E) hMoDCs, treated as in (A–B) and challenged with *T. gondii* (PRU-RFP). X- and y-axes indicate distances in μm. (D, F) Histograms of accumulated

*Figure 4 continued on next page*

*Figure 4 continued*
distances migrated (µm) by (**D**) mBMDCs and (**F**) hMoDCs as in (**C and E**), respectively. Dotted lines indicate median values (n = 3–4 independent experiments). (**G**) Box-and-whisker dot plots show, for each indicated condition, median velocities (µm/min) of unchallenged and *T. gondii*-infected mBMDCs and hMoDCs (n = 3–4 independent experiments). Statistical significance was tested by ordinary one-way ANOVA with Dunnett's multiple comparison test for (**A, B, G**) and by Mann-Whitney test for (**D, F**), *p<0.05, **p<0.01, ***p<0.001, ns p≥0.05.
The online version of this article includes the following figure supplement(s) for figure 4:

**Figure supplement 1.** ShRNA transduction efficiency.

concentrations (*Figure 5F*) in the supernatants of hMoDCs, as previously shown in mBMDCs (*Fuks et al., 2012*). Further, exogenous GABA and GABA-A R agonism (muscimol) fully reconstituted hypermotility in hMoDCs in presence of GABA synthesis inhibitor (*Figure 5E, H*). This, together with the identification of GAD67 as putative principal GABA synthesis enzyme in hMoDCs (*Figure 1G*) motivated gene silencing of GAD67 (*Figure 5I*, *Figure 4—figure supplement 1B,D*, *Supplementary file 3*). Importantly, the amounts of secreted GABA in the supernatant were strongly reduced in GAD67-silenced infected hMoDCs, approached amounts secreted by unchallenged cells and contrasted with maintained GABA secretion by mock- or shLuc-transduced infected cells (*Figure 5J*). Finally, GAD67-silenced infected hMoDCs exhibited abrogated hypermotility (*Figure 5K*). Jointly, these data demonstrate a critical dependence of the hypermigratory phenotype on GABA synthesized by GAD67 in hMoDCs.

## The GABA signaling regulator NKCC1 impacts hypermotility of phagocytes

GABA-A R function in the CNS is regulated by CCCs, which are subdivided in Na-K-Cl cotransporters (NKCCs) and K-Cl cotransporters (KCCs) (*Kaila et al., 2014*). However, CCCs have remained uncharacterized in phagocytes. A transcriptional expression screen (*Supplementary file 1*) detected mRNAs of NKCCs and KCCs in mBMDCs (*Figure 6A*) and in hMoDCs (*Figure 6B*). Notably, upon *T. gondii* challenge, a strong upregulation of NKCC1 was observed in mBMDCs, which was less accentuated in hMoDCs (*Figure 6C,D*). Immunolabeling and western blotting indicated presence of NKCC1/2 proteins in mBMDCs (*Figure 6E,F*). Interestingly, antagonism with bumetanide, at concentrations known to inhibit NKCC but not KCC activity in neurons (*Orlov et al., 2015*), resulted in impaired *T. gondii-/N. caninum*-induced hypermotility in mBMDCs (*Figure 6G,I*, *Figure 3—figure supplement 1A,B*), hMoDCs (*Figure 6H,J*), hMDCs and monocytes (*Figure 1—figure supplement 1D,E*, *Figure 3—figure supplement 1C*). To test this further, we applied gene silencing on NKCC1 (*Figure 6K,L*, *Supplementary file 3*) after validation in cell lines (*Figure 4—figure supplement 1A–E*). Importantly, hypermotility was abolished in NKCC1-silenced mBMDCs and hMoDCs (*Figure 6M, N*). Next, we addressed if NKCC1 and GABA-A R functions were interconnected. First, we observed that stimulation with GABA or with the GABA-A R agonist muscimol failed to reconstitute hypermotility of mBMDCs in the presence of the NKCC1 antagonist bumetanide (*Figure 6O*), contrasting with the reconstitution of hypermotility by GABA and muscimol in cells with abrogated GABA synthesis (*Figure 5G*). Second, shNKCC1- or shGAD67-transduced hMoDCs were stimulated with GABA. Upon silencing of GABA synthesis (shGAD67), addition of exogenous GABA reconstituted hypermotility. In contrast, in NKCC1-silenced cells, GABA failed to reconstitute hypermotility (*Figure 6P,Q*, *Figure 6—figure supplement 1*). These data indicate a link between NKCC1 and GABA/GABA-A R function in parasitized phagocytes. Altogether, the data demonstrate a functional implication of the Na-K-Cl cotransporter NKCC1 in *T. gondii-/N. caninum*-induced hypermotility of phagocytes.

## The VDCC Ca$_V$1.3 mediates hypermotility and transient extracellular Ca$^{2+}$ influx into DCs in response to GABA

We recently reported VDCC expression by murine DCs (*Kanatani et al., 2017*). To address the roles of VDCCs in human DCs, we challenged hMoDCs and hMDCs with *T. gondii*. First, we detected mRNA expression of 9 out of 10 known VDCC subtypes in hMoDCs, with highest relative expression of Ca$_V$1.3, 1.4 and 3.1 (*Figure 7A*, *Supplementary file 1*, *2*). Moreover, challenge with *T. gondii* led to a prominent upregulation of Ca$_V$1.3, 2.2 and 2.3 mRNAs (*Figure 7B*). This result motivated

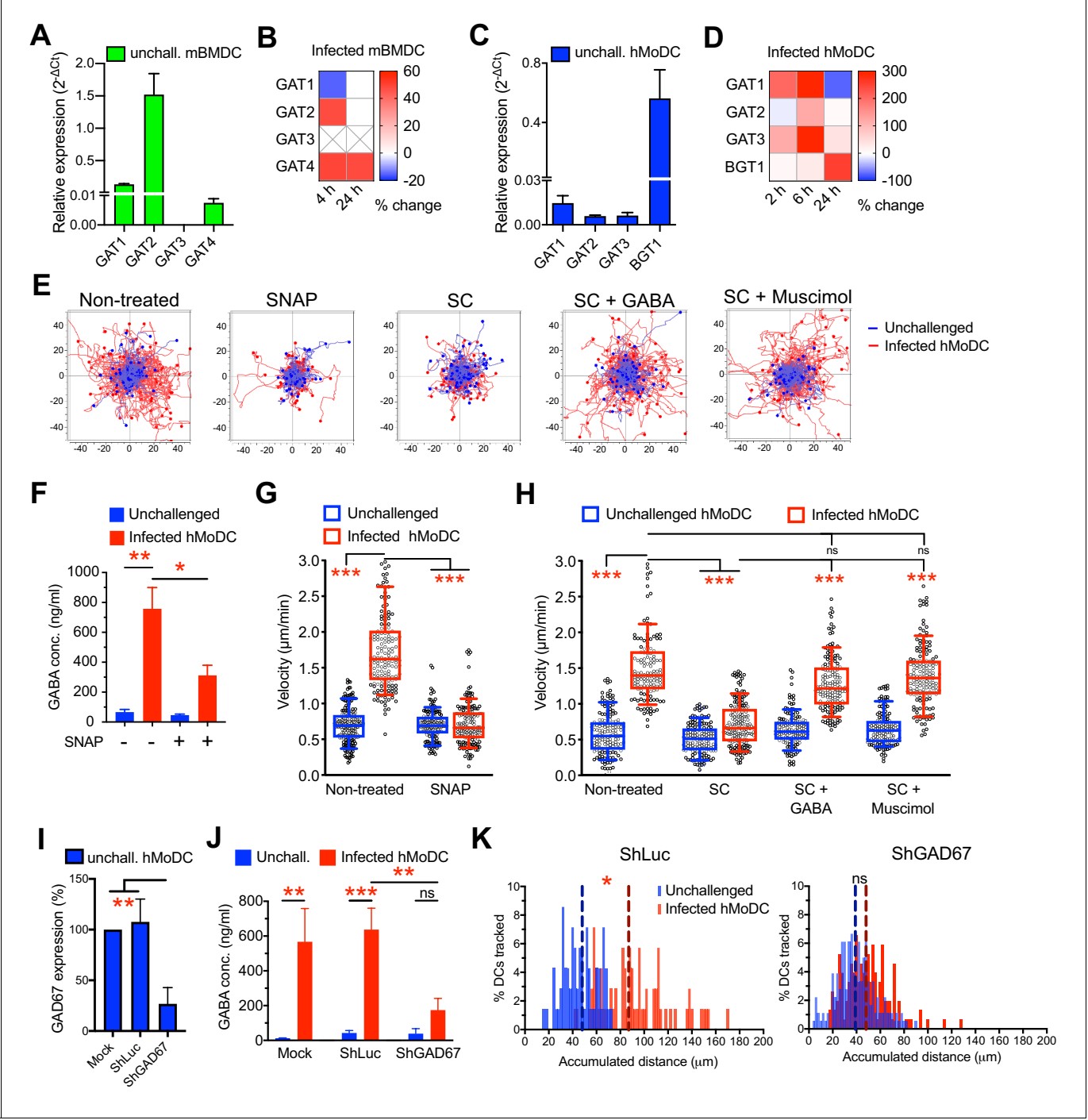

**Figure 5.** Impact of GABA synthesis enzymes and GABA transporters on hypermotility. (A, C) Relative mRNA expression ($2^{-\Delta Ct}$) of GABA transporters in unchallenged (A) mBMDCs and (C) hMoDCs (n = 3 independent experiments). (B, D) Heat maps show (%) transcriptional changes in the expression of GABA transporters upon challenge of (B) mBMDCs and (D) hMoDCs with *T. gondii* (PRU-RFP) relative to unchallenged cells at indicated time-points. (X) indicates no amplification (n = 3–4 independent experiments). (E) Representative motility plots of unchallenged and *T. gondii*-infected hMoDCs treated with GABA transporter inhibitor (SNAP) and GABA synthesis inhibitor (SC) in absence and presence of GABA or GABA analog (muscimol). X- and y-axes indicate distances in µm. (F) GABA (ng/ml) secreted in supernatants of hMoDCs challenged with *T. gondii* (ME49/PTG) in presence of SNAP was quantified by ELISA (n = 3 independent experiments). (G, H) Box-and-whisker dot plots show, for each indicated condition, velocities (µm/min) of unchallenged and *T. gondii*-infected hMoDCs, as in (E) (n = 3 independent experiments). (I) The mRNA expression ($2^{-\Delta Ct}$) in control shLuc-treated and shGAD67-treated unchallenged hMoDCs related (%) to mock-treated cells (n = 5 independent experiments). (J) GABA secreted (ng/ml) in supernatants of mock-, shLuc- and shGAD67-treated hMoDCs challenged with *T. gondii* (PRU-RFP) (n = 5 independent experiments). (K) Histograms show

*Figure 5 continued on next page*

Figure 5 continued

accumulated distances migrated (µm) by control shLuc-treated and shGAD67-treated hMoDCs, respectively, challenged with *T. gondii* (PRU-RFP). Dotted lines indicate median values (n = 3–4 independent experiments). Bar graphs show mean + SEM. Statistical significance was tested by paired t-test for (F), ordinary one-way ANOVA with Dunnett's multiple comparison test for (G, H, I, J) and Mann-Whitney test for (K), *p<0.05, **p<0.01, ***p<0.001, ns p>0.05.

pharmacological inhibition, including broad VDCC inhibition (benidipine), inhibition of $Ca_V1$ subtypes/L-type VDCCs (nifedipine) and targeted inhibition of $Ca_V1.3$ (CPCPT) (*Yao et al., 2006*; *Kang et al., 2012*). Importantly, VDCC inhibition abolished *T. gondii-/N. caninum*-induced hypermotility in hMoDCs (*Figure 7C,D*), hMDCs, monocytes and mBMDCs (*Figure 1—figure supplement 1D,E*, *Figure 3—figure supplement 1A,B,C*). Moreover, a structural analogue of nifedipine with positive ionotropic effect (Bay K8644) reconstituted hypermotility of infected hMoDCs in the presence of GABA synthesis inhibitor (SC) (*Figure 7E*) and also significantly enhanced motility in unchallenged hMoDCs (*Figure 7D*). The prominent effect of $Ca_V1.3$ inhibition prompted us to silence $Ca_V1.3$ expression (*Figure 7F*, *Figure 4—figure supplement 1B,D*, *Supplementary file 3*), which yielded abolished hypermotility (*Figure 7G,H*).

To test the implications of GABAergic signaling on $Ca^{2+}$ responses, $Ca^{2+}$ indicator dye-loaded mBMDCs were challenged with agonistic and antagonistic stimuli and $Ca^{2+}$ responses were measured (*Figure 8A*, *Video 1*). Importantly, perfusion of exogenous GABA consistently generated transient cytosolic $Ca^{2+}$ elevations, which were antagonized by the GABA-A R blocker picrotoxin (broad inhibitor) and with maintained responsiveness by purinergic $Ca^{2+}$ channels to ATP (*Figure 8B,C,D*; *Video 1*). Further, transients generated by L-type VDCC agonism (Bay K) were effectively antagonized by $Ca_V1.3$ inhibition (CPCPT), indicative of a prominent role for $Ca_V1.3$ in the measured $Ca^{2+}$ responses (*Figure 8E,F,G*, *Figure 8—figure supplement 1*). Finally, *T. gondii*-infected cells responded to GABA with transient $Ca^{2+}$ influx. In individual cell recordings, $Ca^{2+}$ influx was significantly reduced or abolished by application of inhibitors of β-subunit containing GABA-A Rs (SCS) and by picrotoxin, with maintained responses to ATP (*Figure 8H,I,J*). Jointly, these data identify a determinant role for GABA-A Rs in the GABA-induced $Ca^{2+}$ influx via $Ca_V1.3$ in DC hypermotility.

## Gene silencing and pharmacological antagonism of GABAergic signaling slow DC migration and reduce parasite loads in mice

To address the impact of GABAergic signaling on DC-mediated parasite dissemination, we designed separate approaches that targeted GABA-A R function or the function of the GABA signaling regulator NKCC1 in mice. First, we controlled that inhibitors had non-significant effects on parasite invasion and replication (*Figure 9—figure supplement 1A*) and a persistent inhibitory effect on hypermotility (*Figure 9—figure supplement 1B*). Second, inhibitor-pretreated parasitized mBMDCs (CMTMR-labeled) and non-treated parasitized mBMDCs (CMF2HC-labeled) were simultaneously adoptively transferred to mice in a competition assay (*Figure 9A*). Fourteen to 18 h post-inoculation, organs were harvested and cells were characterized by flow cytometry (*Figure 9B,C*, *Figure 9—figure supplement 2A, B,C*). Importantly, the ratio of pretreated vs non-treated parasitized mBMDCs was significantly lower in spleen for both GABA-A R inhibitor and NKCC1 inhibitor treatments compared with that for non-treated condition (*Figure 9D*). In contrast, non-significant differences were observed in peritoneum (*Figure 9D*). This indicated selective reduced migration of mBMDCs pretreated with GABA-A R inhibitors or NKCC1 inhibitor compared with non-treated mBMDCs in individual mice. Third, to asses if the reduced migration of parasitized mBMDCs impacted the infection, parasite loads were assessed by plaquing assays and by qPCR. Importantly, compared with the non-treated condition, treatments significantly decreased the parasite loads in spleen and liver (*Figure 9E*, *Figure 9—figure supplement 2D, E*). Finally, we silenced the GABA-A R subunit ρ1 or NKCC1, two GABA signaling targets with a significant impact on mBMDC migration *in vitro* (*Figure 4*, *Figure 6*). When gene-silenced mBMDCs challenged with *T. gondii* were adoptively transferred into mice, significantly reduced parasite loads were quantified in peripheral organs (*Figure 9F*), corroborating results of pharmacological treatments (*Figure 9E*). Moreover, pharmacological inhibition of GABA-A Rs or NKCC1 yielded a significant reduction of parasites loads in the brain by day seven post-inoculation (*Figure 9G*). The data demonstrate a role for the GABA-A R subunit ρ1 and the GABA signaling regulator NKCC1 in mBMDC-mediated dissemination of *T.*

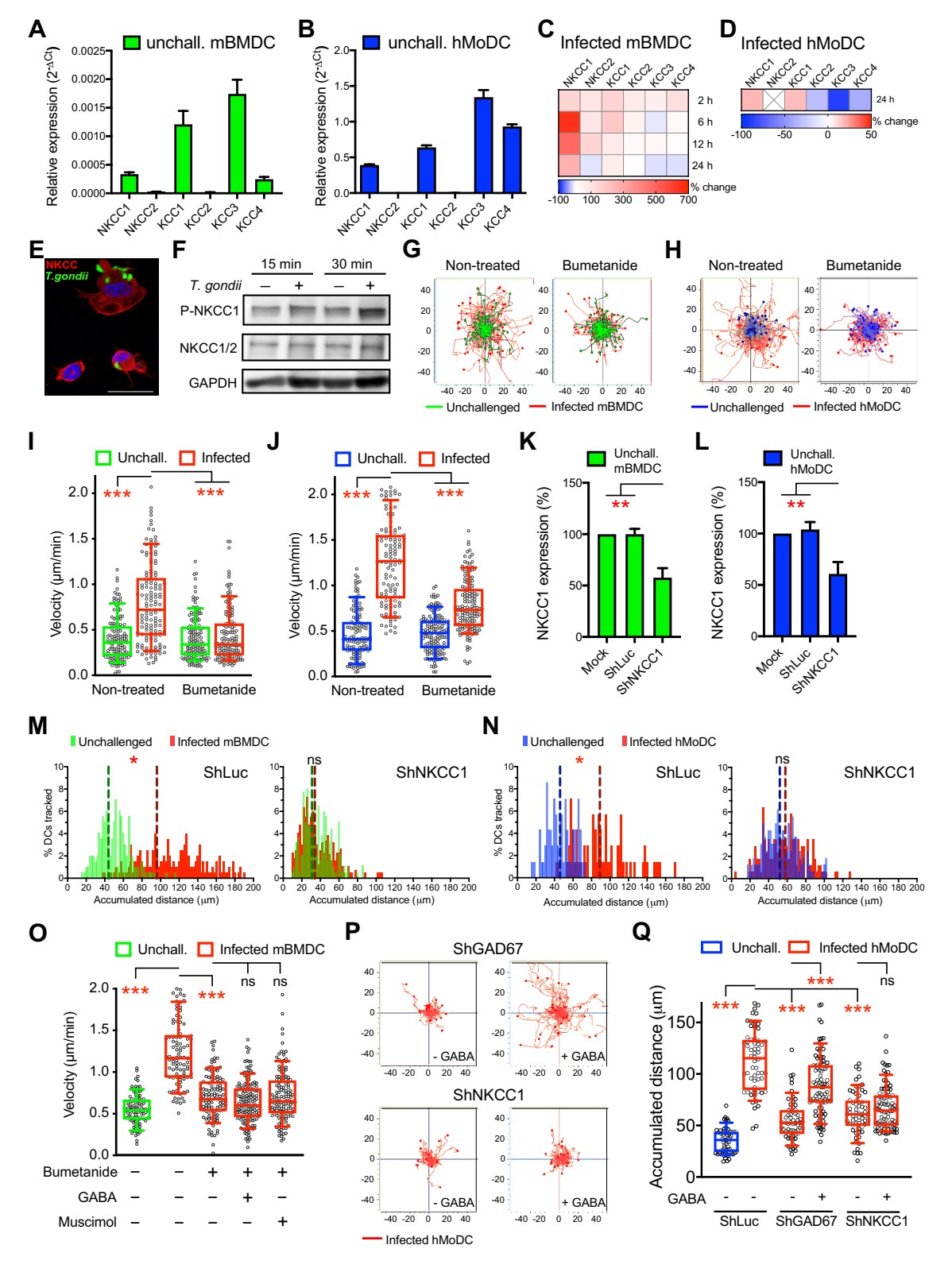

**Figure 6.** NKCC1 is a determinant of GABAergic hypermotility. (A, B) Relative mRNA expression ($2^{-\Delta Ct}$) of cation chloride transporters (CCCs) in unchallenged (A) mBMDCs and (B) hMoDCs (n = 3 independent experiments). (C, D) Heat map depicts (%) transcriptional expression changes of CCCs in (C) mBMDCs and (D) hMoDCs challenged with *T. gondii* (PRU-RFP) relative to unchallenged cells at indicated time points. (X) indicates no amplification (n = 3 independent experiments). (E) Immunostaining of mBMDCs challenged with *T. gondii* (ME49/PTG-GFP) stained with NKCC1/2

*Figure 6 continued on next page*

*Figure 6 continued*

monoclonal antibody (Alexa Flour 594-conjugated anti-mouse secondary antibody) and DAPI (nuclei). Scale bars: 10 µm. (**F**) Representative Western blot of lysates from *T. gondii*-challenged mBMDCs for indicated time, immunoblotted with phospho-NKCC1 and total NKCC1/2 antibodies. GAPDH was used as loading reference. (n = 4 independent experiments). (**G, H**) Representative motility plots of unchallenged and *T. gondii*-infected (**G**) mBMDCs and (**H**) hMoDCs treated with NKCC1 inhibitor (bumetanide). X- and y-axes indicate distances in µm. (**I, J**) Box-and-whisker dot plots show, for each indicated condition, median velocities (µm/min) of unchallenged and *T. gondii*-infected (**I**) mBMDCs and (**J**) hMoDCs as in (**G, H**) (n = 3 independent experiments). (**K, L**) The mRNA expression ($2^{-\Delta Ct}$) of control shLuc- and shNKCC1-treated unchallenged (**K**) mBMDCs and (**L**) hMoDCs related (%) to mock-treated cells (n = 7 independent experiments for mBMDCs and n = 4 for hMoDCs). Bar graphs show mean + SEM. (**M, N**) Histograms show accumulated distances migrated (µm) by control shLuc-treated and shNKCC1-treated (**M**) mBMDCs and (**N**) hMoDCs, respectively, challenged with *T. gondii* (PRU-RFP). Dotted lines indicate median values (n = 3 independent experiments). (**O**) Velocities of unchallenged and *T. gondii*-infected mBMDCs treated with GABA or muscimol in presence of bumetanide. (**P**) Representative motility plots of *T. gondii*-infected shGAD67- and shNKCC1-treated hMoDCs in presence of GABA. (**Q**) Velocities of unchallenged and *T. gondii*-infected shLuc-, shGAD67- and shNKCC1-treated hMoDCs with or without GABA. Statistical significance was tested by ordinary one-way ANOVA with Dunnett's multiple comparison test for (I, J, K, L, O, Q) and by Mann-Whitney test for (M, N), *p<0.05, **p<0.01, ***p<0.001, ns p≥0.05.
The online version of this article includes the following figure supplement(s) for figure 6:

**Figure supplement 1.** Histograms of accumulated distances migrated (µm) by mock- or shLuc-treated unchallenged and *T. gondii*-infected hMoDCs (upper panel), *T. gondii*-infected shGAD67- or shNKCC1-treated hMoDCs with or without GABA (lower panel).

---

*gondii*. Altogether, the findings support the notion that GABAergic signaling promotes the dissemination of *T. gondii* via parasitized DCs.

## Discussion

Signaling pathways that can drive migration of immune cells, and are alternative to canonical chemokine-mediated migration, have remained poorly understood. Here, we establish that human and murine mononuclear phagocytes possess a conserved GABAergic system that, upon activation, promotes migration *in vitro* and *in vivo*. We identified and performed functional tests on the five principal components of GABAergic signaling, namely (*i*) GABA metabolism, (*ii*) GABA transportation and secretion, (*iii*) GABA-A R activation, (*iv*) GABA signaling regulators CCCs and (*v*) effector $Ca^{2+}$ channel signaling by VDCCs (*Figure 10*). The data provide a molecular and cellular framework for assessing the role of the GABAergic system in immune cells.

We demonstrate that a conserved expression of GABAergic molecular components is functionally linked to motility and migratory activation of mononuclear phagocytes upon infection challenge. First, our studies identify GAD67 as the principal GABA synthesizing enzyme in phagocytes. Gene silencing and pharmacological inhibition of GAD67 abrogated secretion of GABA and migratory activation of phagocytes, which was reconstituted by GABA-A R agonism. This supports the notion that GABA is synthesized cytosolically and secreted in vesicle-independent fashion, likely by transport through GATs, for tonic modulations of GABA-A Rs in immune cells, similar to neurons (*Kaufman et al., 1991*; *Feldblum et al., 1993*). Second, our data establish that expression of specific GABA-A R subunits determine the motogenic function of GABA-A Rs in phagocytes. The expression of GABA-A R subunit types was diverse, in line with the expression diversity in neurons (*Davis et al., 2000*; *Goetz et al., 2007*). Yet, the different phagocyte types consistently expressed a repertoire of GABA-A R subunits sufficient to constitute functional channels, that is: at least one α, one β and one third type of subunit, or homopentamer-forming ρ subunits. The inhibitory effects by selective pharmacological antagonism on phagocyte hypermotility indicated implication of α, β and ρ subunits and was confirmed by gene silencing. Importantly, silencing of the α4 subunit (but not ρ2) or β3 and ρ1 subunits (but not α3) abolished hypermigration of human and murine DCs, respectively. Jointly, this narrows putative receptor pentamers acting in phagocytes but also highlights a hierarchy among GABA-A R subunits mediating migratory activation or function redundancy among the different subunits. Third, our data identify a determinant role for the CCC NKCC1 in the migratory activation of phagocytes. By pharmacological inhibition and gene silencing *in vitro* and *in vivo*, we show that NKCC1 plays a crucial role in the regulation of GABA signaling in phagocytes. Of note, NKCC1 was linked to GABA-A R function and its regulative characteristics together with limited variability -compared with the high diversity in expression of GABA-A R subunits- made NKCC1 a prime target for *in vivo* experimentation. Finally, we demonstrate that stimulation with GABA elicits $Ca^{2+}$ influx

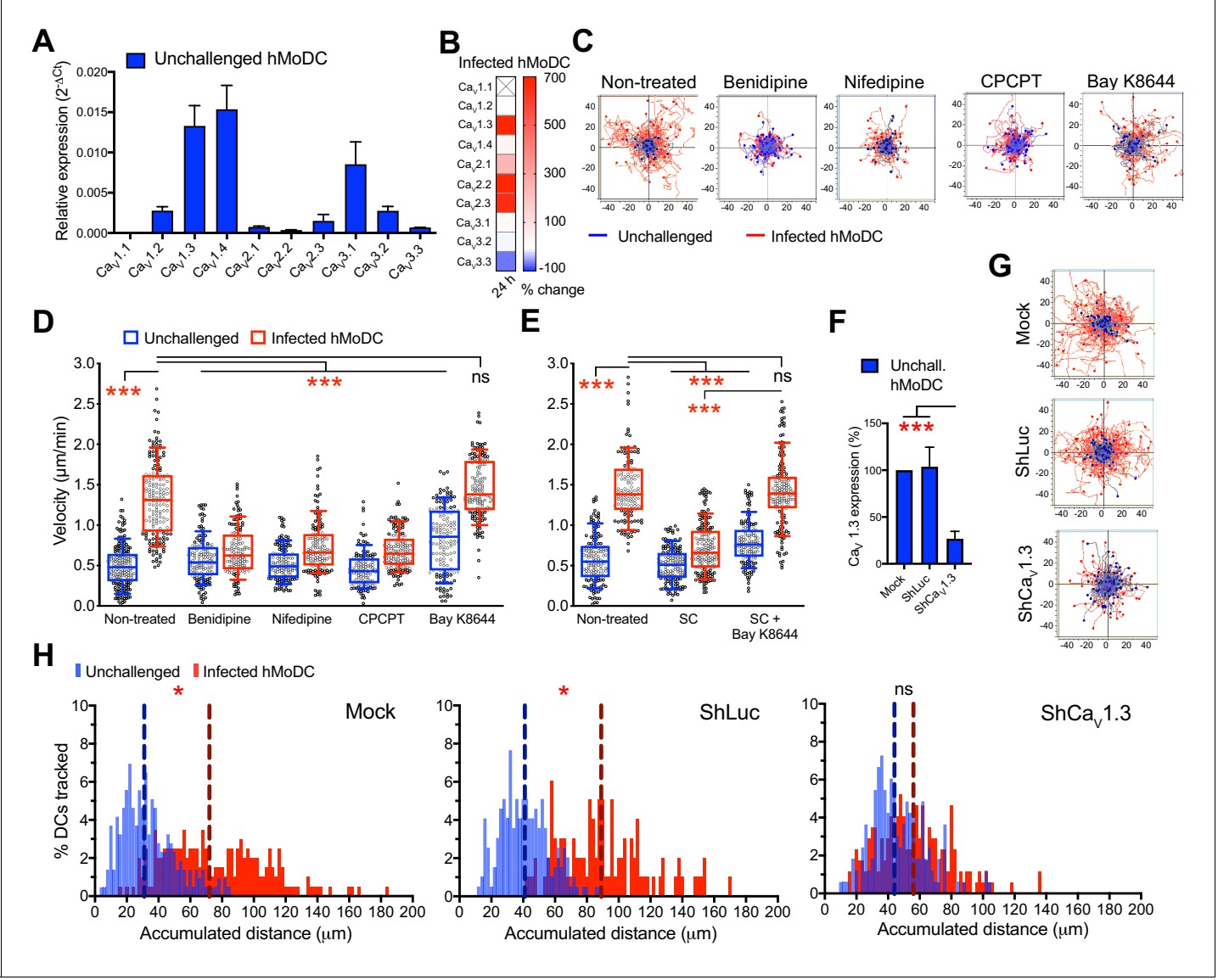

**Figure 7.** $Ca_V1.3$ is the mediator for downstream effects of activated GABA signaling. (**A**) Relative mRNA expression ($2^{-\Delta Ct}$) of voltage-dependent calcium channels (VDCCs) in unchallenged hMoDCs (n = 3 independent experiments). (**B**) Heat map shows (%) transcriptional changes in the expression of VDCCs subunits upon challenge of hMoDCs with *T. gondii* (PRU-RFP) relative to unchallenged cells at indicated time-point. (**X**) indicates no amplification (n = 3 independent experiments). (**C**) Representative motility plots of unchallenged and *T. gondii*-infected hMoDCs treated with benidipine (VDCC broad inhibitor), nifedipine (L-type inhibitor), CPCPT ($Ca_V1.3$ specific inhibitor) and bay K8644 (VDCC activator). X- and y-axis are distances in μm. (**D**) Box-and-whisker dot plots show, for each indicated condition, velocities (μm/min) of unchallenged and *T. gondii*-infected hMoDCs, as in (**C**), and (**E**) treated with SC (GABA synthesis inhibitor) in presence of bay K8644 (n = 3 independent experiments). (**F**) The mRNA expression in control shLuc- and shCa$_V$1.3-treated unchallenged hMoDCs related (%) to mock-treated cells (n = 10 independent experiments). (**G**) Representative motility plots of hMoDCs, treated as in (**F**) and challenged with *T. gondii* (PRU-RFP). (**H**) Histograms of accumulated distances migrated (μm) by hMoDCs as in (**G**). Dotted lines indicate median values (n = 3 independent experiments). Bar graphs show mean + SEM. Statistical significance was tested by ordinary one-way ANOVA with Dunnett's multiple comparison test for (**D, E, F**) and by Mann-Whitney test for (**H**), *p<0.05, ***p<0.001, ns p≥0.05.

transients in the DC cytosol. In line with this, human and murine phagocytes expressed a highly conserved repertoire of VDCC subtypes. Moreover, silencing of the VDCC subtype $Ca_V1.3$ in human DCs abrogated *T. gondii*-induced hypermotility, in line with our observations in murine DCs (*Kanatani et al., 2017*). Jointly, this demonstrates that $Ca_V1.3$ is determinant to the motogenic action of GABA-A R activation.

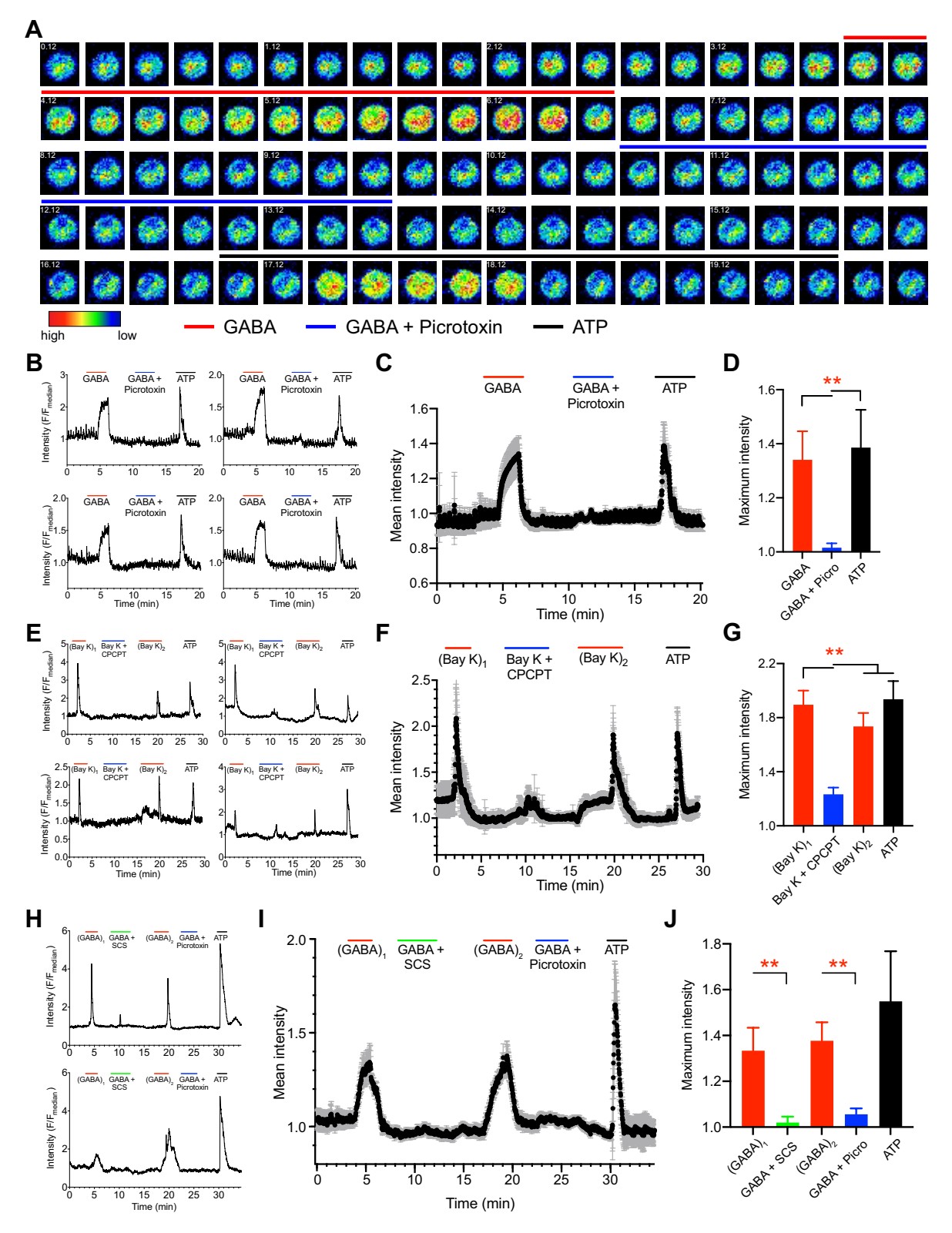

**Figure 8.** Ca²⁺ responses in unchallenged and *T. gondii*-infected DCs upon agonism and antagonism of GABA-A Rs and VDCCs. (**A**) Representative time-lapse micrographs show live cell Ca²⁺ imaging of one single mBMDC preloaded with Ca²⁺ indicator dye (Fluo-8H AM) and sequentially perfused with GABA (1 mM), GABA (1 mM) + picrotoxin (100 µM) and ATP (150 µM). Colored lines indicate perfusion times (min:s) of treatments. Color scale depicts relative fluorescence intensity. (**B**) Representative live cell Ca²⁺ recording traces (relative fluorescence intensity, F/F_median) from four cells plotted

*Figure 8 continued on next page*

*Figure 8 continued*

against time (min) as in (**A**). Lines indicate respective perfusions times. (**C**) Mean $Ca^{2+}$ response intensity (black dots) and SEM (gray whiskers) from one recording with 105 cells, plotted against time as in (**B**), and (**D**) maximum $Ca^{2+}$ response intensity of 294 cells from three independent experiments. (**E**) Representative live cell $Ca^{2+}$ recording traces from four cells sequentially perfused with bay K 8644 (40μM), bay K 8644 (40μM) + CPCPT (100 μM), 2nd application of bay K 8644 (40μM) and ATP (150 μM) at times indicated by lines. (**F**) Mean $Ca^{2+}$ response intensity (black dots) and SEM (gray whiskers), from 72 cells plotted against time as in (**E**), and (**G**) maximum $Ca^{2+}$ response intensity of 233 cells from three independent experiments. (**H**) Representative live cell $Ca^{2+}$ recording traces from *T. gondii* (ME49-RFP)-infected cells sequentially perfused with GABA (1 mM), GABA (1 mM) + SCS (100 μM), GABA (1 mM), GABA (1 mM) + picrotoxin (100 μM) and ATP (150 μM) at times indicated by lines. (**I**) Mean $Ca^{2+}$ response intensity (black dots) and SEM (gray whiskers) from 44 cells, plotted against time as in (**H**), and (**J**) maximum $Ca^{2+}$ response intensity of 115 cells from three independent experiments. Bar graphs show mean + SEM. Statistical significance was assessed by ordinary one-way ANOVA with Dunnett's multiple comparison test for (**D, G, J**), **$p < 0.01$.

The online version of this article includes the following figure supplement(s) for figure 8:

**Figure supplement 1.** Single-cell micrographs of $Ca^{2+}$ influx show VDCC activation.

Our data establish that *T. gondii* and *N. caninum*, two coccidian parasites with a broad range of vertebrate hosts, induce GABAergic signaling in parasitized phagocytes. From a perspective of intracellular parasitism, hijacking a conserved GABAergic system offers the advantage of inducing migratory activation of shuttle phagocytes within minutes after active invasion (*Weidner et al., 2013*) to favor systemic dissemination in vertebrate hosts (*Lambert et al., 2006*; *Courret et al., 2006*). Activation of GABAergic signaling is relatively fast as GABA binding opens the GABA-A Rs in milliseconds (*Farrant and Nusser, 2005*) and the elements that synthesize and transport GABA, GADs and GATs, respectively, are constitutionally expressed and upregulated upon infection. Consequently, DCs initiated GABA secretion shortly after *T. gondii* invasion. Thus, the rapid onset and tight regulation of GABAergic signaling by-passes the need for, presumably slower, transcriptional regulation. Additionally, the GABA signaling regulator NKCC1 plays an important role in the hypermigration of parasitized phagocytes, presumably by impacting chloride concentration and thus, GABA-A R function. Consequently, inhibition of GABA-A Rs or NKCC1 in adoptively transferred pharmacologically pretreated or gene silenced DCs, significantly reduced the migration of parasitized DCs and parasite loads in mice. Importantly, hampered dissemination to peripheral vital organs was evident early during infection and, later, resulted in reduced parasite loads in the CNS. Jointly with present data, different approaches show that targeting (*i*) GABA synthesis/transportation (*Fuks et al., 2012*), (*ii*) GABA-A R signaling, (*iii*) GABA-A R regulation/NKCC1 hampers systemic dissemination and parasite loads in the brain. However, inhibition of VDCC signaling inhibited systemic dissemination but non-significantly impacted parasite loads in the brain (*Kanatani et al., 2017*), indicating that specific GABAergic signaling components contribute differently or indicating redundancy in VDCC signaling. Because invasive coccidian parasites need to reconcile their obligate intracellular existence with the need for dissemination, the hijacking of migratory leukocytes represents a secluded replication niche that facilitates dissemination. The identified GABAergic determinants provide a molecular framework for assessing if other protozoa, intracellular bacteria (*Kim et al., 2018*) or viruses (*Zhu et al., 2017*) utilize the GABAergic signaling of phagocytes for dissemination. Because hypermigratory parasitized DCs

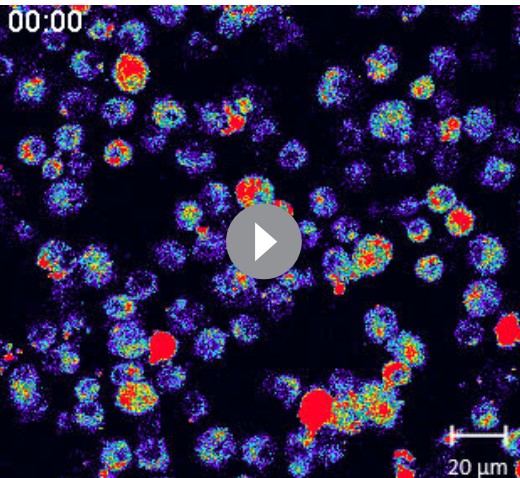

**Video 1.** Representative time-lapsed micrographs from a live cell. $Ca^{2+}$ imaging recording of mBMDC preloaded with $Ca^{2+}$ indicator dye (Fluo-8H AM) and sequentially perfused with GABA (1 mM), GABA (1 mM) + picrotoxin (100 μM) and ATP (150 μM) at 3.42–6.46, 10.35–13.32 and 17.08–19.52 min, respectively. The Fluo-8H AM fluorescence signal (green) is converted to rainbow color scale depicting relative fluorescence intensity ranging from blue (lowest) to red (highest). Scale bar 20 μm.
https://elifesciences.org/articles/60528#video1

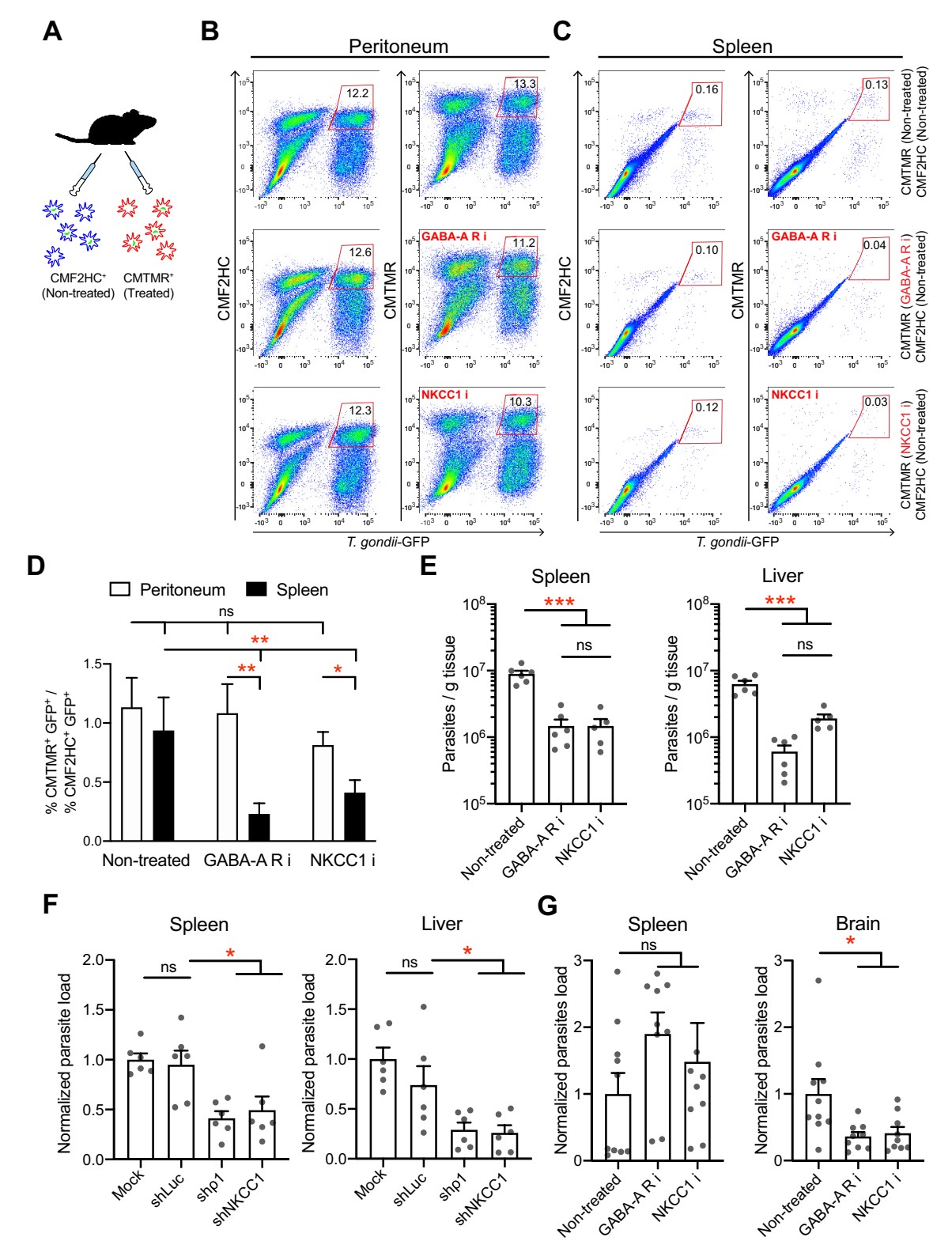

**Figure 9.** *In vivo* impact of GABAergic inhibition on DC migration and parasite loads. (**A**) Schematic illustration of simultaneous adoptive transfers of *T. gondii* (ME49/PTG-GFP)-challenged mBMDCs prelabeled with CMF2HC or CMTMR dye and pretreated as described in Materials and methods, respectively. (**B, C**) Representative bivariate plots show CD11c⁺ cells from (**B**) peritoneal cavity and (**C**) spleen of C57BL/6 mice inoculated with *T. gondii*-challenged mBMDCs prelabeled with CMF2HC or CMTMR dye. Cells were analyzed by flow cytometry at 14–18 h post-inoculation (gating

*Figure 9 continued on next page*

*Figure 9 continued*

strategy in *Figure 9—figure supplement 2A, B,C*). Plots in upper, center and lower rows show, respectively, non-treated cells, cells pretreated with GABA-A R inhibitors (GABA-A R i) and NKCC1 inhibitor (NKCC1 i). (D) Bar graph shows the ratio of treated cells (CD11c⁺GFP⁺CMTMR⁺) to non-treated cells (CD11c⁺GFP⁺CMF2HC⁺) in peritoneal lavage and spleen after adoptive transfer of *T. gondii*-challenged mBMDCs (n = 7–10 mice per group). (E) Parasite loads in spleen and liver of C57BL/6 mice at day 4 post-inoculation of pharmacologically treated cells (GABA-A R i, NKCC1 i) related to non-treated cells and measured by plaquing assays (n = 5–6 mice per group). (F) Parasite loads in spleen and liver of C57BL/6 mice at day 5 post-inoculation of gene-silenced cells (shρ1, shNKCC1) related to mock- and control shLuc-transduced cells and measured by plaquing assays (n = 6 mice per group). (G) Parasite loads in spleen and brain of CD1 mice at day 7 post-inoculation of pharmacologically treated cells (GABA-A R i, NKCC1 i) related to non-treated cells and measured by plaquing assays (n = 9–10 mice per group). Bar graphs show mean + SEM. Statistical significance was assessed by ordinary one-way ANOVA with Tukey's multiple comparison test, *p<0.05, **p<0.01, ***p<0.001, ns p≥0.05.

The online version of this article includes the following figure supplement(s) for figure 9:

**Figure supplement 1.** Effect of GABAergic inhibition on parasite replication in DCs.
**Figure supplement 2.** Gating strategy for flow cytometry analysis.

exhibit chemotaxis (*Weidner and Barragan, 2014*) and GABA impacts the secretion of pro-inflammatory cytokines (*Bhandage et al., 2018*), the impact of GABAergic signaling in the inflammatory microenvironment of infection needs to be further investigated, also in the setting of acute and chronic infection and neuroinflammatory responses in the CNS (*Bhandage et al., 2019*). Thus, hypermigration and chemotaxis are not antithetical and may, in fact, cooperatively potentiate the migratory potential of parasitized phagocytes and therefore also the dissemination of coccidia (*Fuks et al., 2012*; *Weidner et al., 2013*; *García-Sánchez et al., 2019*).

Our study provides the first exhaustive characterization of a GABAergic machinery in myeloid mononuclear phagocytes. Recent findings also indicate GABAergic responses by macrophages (*Januzi et al., 2018*), microglia (*Bhandage et al., 2019*), lymphocytes (*Bhandage et al., 2018*) and bovine immune cells (*García-Sánchez et al., 2019*). Altogether, this highlights that GABAergic signaling by immune cells may be more the rule than the exception. Along these lines, GABA has a motogenic role in embryonic interneuron migration in the developing fetus (*Bortone and Polleux, 2009*). Furthermore, GABAergic signaling has newly been linked to the metastasis of multiple cancer types (*Sizemore et al., 2014*; *Wu et al., 2014*), including gliomas, pancreatic cancer and breast cancer (*Neman et al., 2014*; *Takehara et al., 2007*; *Smits et al., 2012*). The peripheral GABAergic system also appears implicated in various autoimmune diseases, such as multiple sclerosis (*Bhat et al., 2010*), type I diabetes (*Li et al., 2017*; *Bhandage et al., 2018*) and rheumatoid arthritis (*Tian et al., 2011*), where GABAergic inhibition dampens the inflammatory response. It will be important to assess if the motogenic molecular components identified here are also implicated in the inflammatory responses and in cancer cell metastasis. Moreover, the patho-physiological cellular microenvironments result in expression of specific subtypes of GABA receptors (*Bhandage et al., 2014*; *Smits et al., 2012*) and thus, selective compounds have been identified in neuropsychiatric drug design (*Korpi and Sinkkonen, 2006*; *Krall et al., 2015*). Additionally, anesthetics targeting GABA-A Rs have been implicated in the impairment of immune cell function during human surgery (*Wheeler et al., 2011*). Thus, receptor subtypes or other GABAergic components may be targeted to modulate migration of GABAergic cells (*Miao et al., 2010*) and our data provide a molecular framework for therapeutic targets of clinical relevance.

Finally, the non-protein amino acid GABA has developed into an essential neurotransmitter of the evolved vertebrate CNS. However, GABA precedes the development of the CNS as a metabolism- and stress-related signaling molecule in prokaryotes, invertebrates and plants (*Hudec et al., 2015*; *MacRae et al., 2012*; *Pinan-Lucarré et al., 2014*). Here, we add that mononuclear phagocytes express a conserved GABAergic system linked to their migratory functions, which coccidian parasites can hijack for dissemination. Thus, GABA also acts as an interspecies signaling molecule in host-microbe interactions.

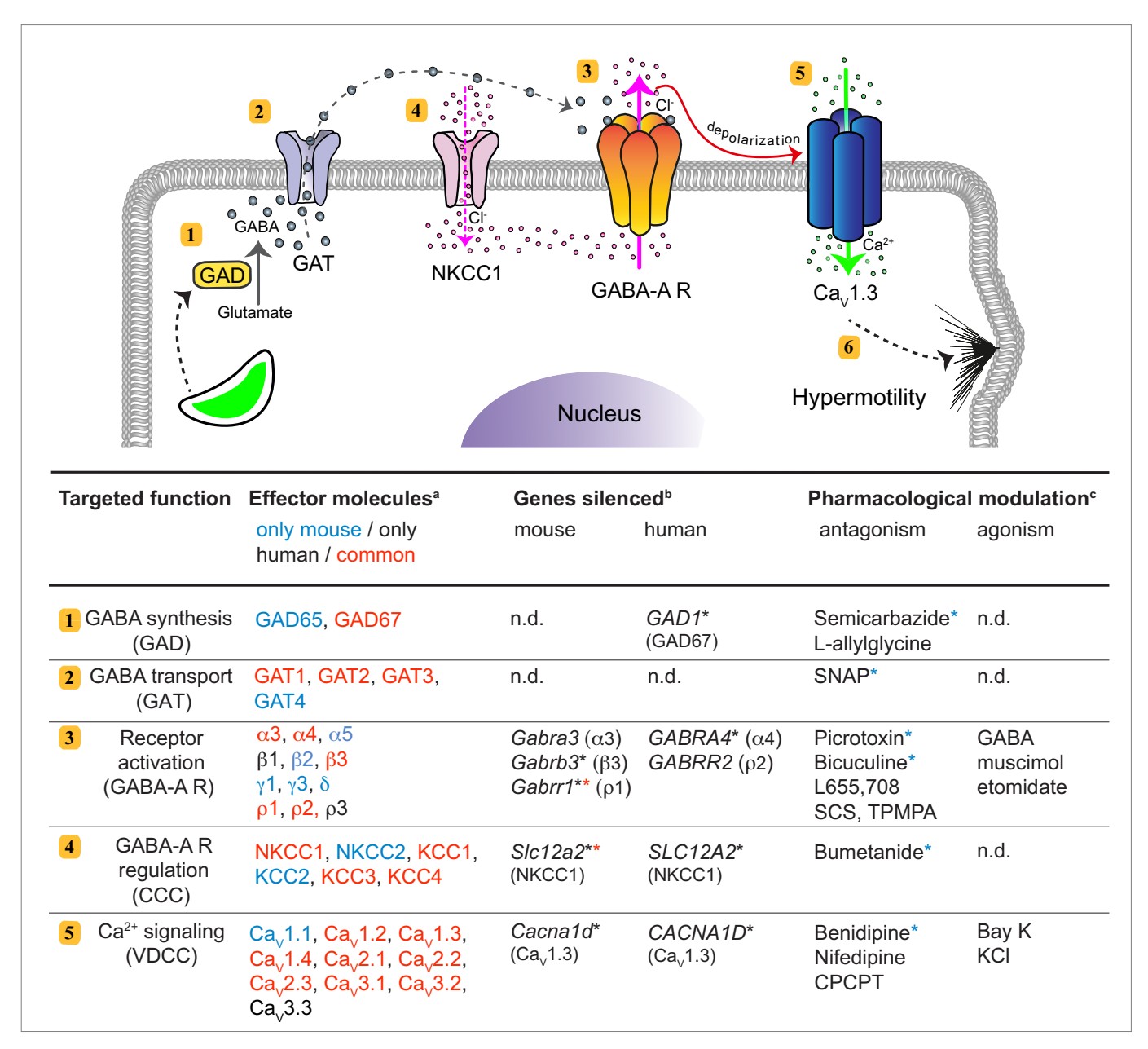

**Figure 10.** GABAergic signaling in mononuclear phagocytes with an impact on cell migration and coccidian parasite dissemination. Schematic representation illustrates the molecular GABAergic signaling components (1-5) identified in human and murine mononuclear phagocytes. Their functions and experimental targeting approaches are detailed in the tabular representation, respectively. (6) $Ca^{2+}$ influx sets the cell in a hypermotile state by activation of MAP kinases and cytoskeletal rearrangements (*Ólafsson et al., 2020*). In the tabular representation, [a] Red colored text indicates molecular components commonly expressed by mouse and human phagocytes. Blue and black color indicates components only detected in mouse and human cells, respectively. [b] Indicates genes targeted by shRNA with an impact on cell migration *in vitro* [*], as described under Materials and methods. [*] Red asterisks indicate conditions additionally tested *in vivo* in mice. [c] Indicates pharmacological agonists and antagonists with an impact on phagocyte motility *in vitro* tested on both mouse and human phagocytes. [*] Blue asterisks indicate conditions additionally tested *in vivo* in mice. n.d.: not determined.

## Materials and methods

### Experimental animals

C57BL/6NCrl mice (6–10 weeks old) and female Crl:CD1(ICR) mice (6–8 weeks old) were purchased from Charles River (Sulzfeld, Germany) and maintained and/or bred under pathogen-free conditions at Experimental Core Facility (ECF), Stockholm University, Sweden.

### Parasites and cell lines

*T. gondii* lines used include GFP-expressing RH (type I), ME49/PTG (type II) and RFP-expressing PRU or ME49 (type II) (*Kim et al., 2001*; *Hitziger et al., 2005*). *N. caninum* lines used include NC-1 and NC-Liverpool (ATCC 50977 and ATCC 50845, American Type Culture Collection, Manassas, Virginia, US). Tachyzoites were maintained by serial 2-day passaging in human foreskin fibroblast (HFF-1, ATCC SCRC-1041) monolayers cultured in DMEM (Thermofisher scientific, Stockholm, Sweden) with 10% fetal bovine serum (FBS; Sigma-Aldrich, Darmstadt, Germany), gentamicin (20 µg/ml; Thermofisher), glutamine (2 mM; Thermofisher), and HEPES (0.01 M; Thermofisher), defined as complete medium (CM). The murine DC cell line JAWSII (ATCC CRL-11904) and neuroectodermal cell lines NE4Cs (ATCC CRL-2925) were cultured in DMEM supplemented with 10% FBS, gentamicin, glutamine, HEPES and human neuronal cell line SH-SY5Y (ATCC CRL-2266) was cultured in Opti-MEM supplemented with 10% FBS and gentamicin. All cultures were regularly tested for mycoplasma.

### Primary cells

Mouse bone marrow-derived DCs (mBMDCs) were generated as previously described (*Fuks et al., 2012*). Briefly, bone marrow cells extracted from legs of 6–10 week-old C57BL/6 mice were cultivated in RPMI 1640 (Thermofisher) with 10% FBS, gentamicin, glutamine and HEPES, additionally supplemented with recombinant mouse GM-CSF (10 ng/ml; Peprotech, Stockholm, Sweden) at 37°C. Medium was replenished on days 2 and 4. Loosely adherent cells harvested on days 6–8 were used for experiments. Human monocytes were isolated from buffy coats, obtained from Karolinska University Hospital, by negative selection using RosetteSep Human Monocyte Enrichment Cocktail (Stemcell Technologies, Cambridge, UK). The population obtained exhibited CD14+ (DakoCytomation, Glostrup, Denmark) and <1% CD3+/19+ (BD Biosciences, Stockholm, Sweden), as evaluated by flow cytometry. For differentiation into hMoDCs, cells were cultivated in DMEM with 10% FBS, gentamicin, glutamine and HEPES, additionally supplemented with recombinant human GM-CSF (75 ng/ml; Peprotech) and IL-4 (30 ng/ml; Peprotech) at 37°C. Medium was replenished on day 3 and day 6. Loosely adherent cells harvested on day 6–9 were used for experiments. DCs were typified by expression of CD1a, CD11b, CD14 (DakoCytomation), CD80, CD83, CD86, HLA-DR, CD11a, CD18, CD54 (BD Biosciences). CD1c+ hMDCs isolated by human Myeloid Dendritic Cell Isolation Kit (Miltenyi Biotec, Bergisch Gladbach, Germany) were 70–97% pure, as characterized by flow cytometry.

### Reagents

Picrotoxin (50 µM), L-655 708 (11,12,13,13a-Tetrahydro-7-methoxy-9-oxo-9H-imidazo[1,5-a]pyrrolo [2,1 c][1,4]benzodiazepine-1-carboxylic acid, ethyl ester, 10 µM), SCS (Salicylidene salicylhydrazide, 1 µM), TPMPA ((1,2,5,6-Tetrahydropyridin-4-yl) methylphosphinic acid, 50 µM), etomidate (10 µM), allopregnanolone (100 nM), muscimol (5 µM), bumetanide (10 µM), nifedipine (10 µM), benidipine (10 µM), bay K 8644 (10 µM), bicuculline (50 µM), ATP (150 µM; all from Tocris Bioscience, Bristol, UK), SNAP ((S)-Nitroso-N-acetylpenicillamine, 50 µM), SC (semicarbazide 50 µM), GABA (5 µM), geneticin (10 µM; all from Sigma-Aldrich) and CPCPT (1-(3-Chlorophenethyl)−3-cyclopentylpyrimidine-2,4,6-(1H,3H,5H)-trione, 1 µM, Merck Millipore, Darmstadt, Germany) were used at the indicated concentrations, if not differently stated. All pharmacological treatments and live cell stainings were performed in CM at 37°C and 5% $CO_2$.

### Motility assays

Cell motility analyses was performed as previously described (*Fuks et al., 2012*; *Weidner et al., 2013*). Briefly, cells were challenged with freshly egressed tachyzoites and subjected to pharmacological treatments for 4–6 h. Cells were seeded in 96-well plates and imaged every min for 60 min

(Zeiss Observer Z.1). Motility tracks for 50–60 cells per treatment were analyzed using ImageJ software for each experiment.

## Immunocytochemistry

mBMDCs were seeded on gelatin-coated glass coverslips for 0.5–1 h, fixed with 4% PFA in PBS for 15–20 min at RT and permeabilized using 0.1% Triton X-100 in PBS. To visualize host cell F-actin, cells were stained with Alexa Fluor 488- or 594-conjugated phalloidin (Invitrogen). To probe GABA-A R subunits, cells were incubated with rabbit anti-GABA-A R α3 polyclonal antibody, rabbit anti-GABA-A R α5 polyclonal antibody, rabbit anti-GABA-A R ρ1 polyclonal antibody (all from Alomone labs, Jerusalem, Israel), mouse anti-GABA-A R β3 monoclonal antibody (NeuroMab, UC Davis, CA, US) and for NKCC, with mouse anti-NKCC1/2 monoclonal antibody (clone T4, Developmental Studies Hybridoma Bank, DSHB, Iowa, USA) ON at 4°C. Following staining with respective Alexa Fluor 488- or 594-conjugated secondary antibodies (Thermofisher) and DAPI, coverslips were mounted and imaged by confocal microscopy (LSM 780 and LSM 800, Zeiss).

## Real-time quantitative PCR

Total RNAs were extracted using Direct-zol miniprep RNA kits (Zymo Research, Irvine, CA, USA) with TRI reagent (Sigma-Aldrich) and first-strand cDNA was synthesized using Superscript IV or Maxima H Minus Reverse Transcriptase (Thermofisher) using a standard protocol. Real-time quantitative PCR (qPCR) was performed in QuantStudio 5 384 Optical well plate system (Applied Biosystem, Stockholm, Sweden) and/or LightCycler 480 (Roche, Basel, Switzerland) in a standard 10 µl with the 2X SYBR FAST qPCR Master Mix (Sigma-Aldrich) with gene-specific primers (*Supplementary file 1*) as described (*Bhandage et al., 2019*). Relative expression ($2^{-\Delta Ct}$) were determined for each target in reference to a normalization factor, geometric mean of reference genes, either importin 8 (IPO8) and TATA-binding protein (TBP) or Actin-β (ACTB) and glyceraldehyde 3-phosphate dehydrogenase (GAPDH).

## GABA enzyme-linked immunosorbent assays

ELISA (Labor Diagnostica Nord, Nordhorn, Germany) was performed as previously described (*Fuks et al., 2012*). Briefly, cells were plated at a density of $1 \times 10^6$ cells per ml and incubated for 24 h, unless differently stated, in presence of *T. gondii* and *N. caninum* tachyzoites. GABA concentrations in supernatants quantified at a wavelength of 450 nm (VMax Kinetic ELISA Microplate Reader, Molecular Devices, Vantaa, Finland).

## Live cell time-lapse calcium imaging

Cells were loaded with, a calcium indicator, 2 µM Fluo-8H AM (AAT Bioquest, Sunnyvale, CA, USA) for 30 min at 37°C and seeded on 5% 3-aminopropyltriethoxysilane (TESPA, Sigma-Aldrich)-coated coverslip for 2–5 min at 37°C. Time lapse images were acquired by confocal microscopy (LSM 800, Zeiss) with 20X or 63X objective at an interval of 1 s per image. Drugs were diluted in RPMI 1640 (without phenol red) and perfused at indicated time and concentration by a peristaltic pump (1 ml/min). Absolute fluorescence intensity values (F) were extracted using ZENBlue software (Zeiss) and relative intensity ($F/F_{median}$) at a given time were analyzed for individual cells. Further, mean intensity of all cells was calculated.

## Western blot

mBMDCs and mouse brain hippocampal tissue were lysed in RIPA buffer with protease and phosphatase inhibitor cocktail (Thermofisher), sonicated, diluted with 4 x laemmli buffer and boiled. Protein were subjected to SDS-PAGE on 8% polyacrylamide gels, transfered onto PVDF membrane (Merck Millipore), blocked in 2.5% BSA and incubated ON with rabbit anti-phospho-NKCC1 polyclonal antibody (Thr212/Thr217, Merck Millipore), mouse anti-NKCC1/2 monoclonal antibody (clone T4, DSHB) and rabbit anti-GAPDH antibody (Merck Millipore) followed by incubation with respective HRP-conjugated secondary antibodies (Cell signaling, Leiden, Netherlands). Protein bands were revealed by enhanced chemiluminescence reagents (Thermofisher) in a ChemiDoc system (BioRad, Stockholm, Sweden).

## Lentiviral vector production and transduction

Self-inactivating Lentiviral particles were produced from Lenti-X 293 T cells (Takara Bio, Gothenburg, Sweden) by co-transfecting (*i*) a lentiviral vector, pLL3.7 or pLKO.1, containing self-complementary hairpin DNA oligos targeting specific mRNA (*Supplementary file 3*), (*ii*) psPAX2 packaging vector and (*iii*) pCMV-VSVg envelope vector, as described previously (*Kanatani et al., 2017*; *Ólafsson et al., 2019*). The lentiviral supernatants were used for transduction. The lentivirus transduction efficiency was examined in murine NE4Cs and human SH-Sy5y cell lines and further, knockdown efficiency was determined in NE4Cs (*Figure 4—figure supplement 1*). Murine BMDCs were transduced on day 3 of culturing, whereas human MoDCs were transduced day 3 and day 5. Transduction efficiency was examined for eGFP expression by epifluorescence microscopy, followed by expression analysis by qPCR for knock-down of targeted mRNA. All the target conditions were compared to mock-condition and a positive control, non-related sequence (luciferase, Luc). Transduced cells were further used for experiments.

## Adoptive transfers of *T. gondii*-infected DCs

Adoptive transfers were performed as previously described (*Kanatani et al., 2017*). Briefly, (*i*) mBMDCs were challenged with freshly egressed tachyzoites (ME49/PTG-GFP, 6 h, MOI 1.5) and 1 h post-challenge, treated with picrotoxin (50 µM) and bicuculline (50 µM), for inhibition of GABA-A R or with bumetanide (40 µM), for inhibition of NKCC1 or (*ii*) Mock-, shLuc, shρ1 and shNKCC1 transduced mBMDCs were challenged with freshly egressed PRU-RFP tachyzoites (6 h, MOI 1.5). Extracellular parasites were removed by centrifugation. Following resuspension, *T. gondii*-infected mBMDCs (equivalent to 50,000 colony-forming units of *T. gondii*) were adoptively transferred into recipient C57BL/6 or Crl:CD1(ICR) mice by an intraperitoneal (i.p.) injection. C57BL/6 mice injected with pharmacologically treated cells or with transduced cells were sacrificed at days 4 and 5 post-infection, respectively. Further, Crl:CD1(ICR) mice injected with pharmacologically treated cells were sacrificed at day 7 post-infection.

For competition assays, mBMDCs were challenged with freshly egressed tachyzoites (ME49/PTG-GFP, MOI 1.5). One h post-challenge, cells were pharmacologically treated with picrotoxin (50 µM) and bicuculline (50 µM) or with bumetanide (40 µM) for 5 h. Further, treated cells were stained with CMTMR (1 µM) and non-treated cells were stained with CMF2HC (2 µM) for 30 min. Cells were washed and adoptively transferred into C57BL/6 mice. Each mouse received two consecutive i.p. injections, first with treated CMTMR-prelabeled cells ($5 \times 10^6$) and second with non-treated CMF2HC-prelabeled cells ($5 \times 10^6$). Mice were sacrificed 14–18 h post-infection to collect organs and cells from peritoneal lavages. The organs were triturated, filtered through 40-µm cell strainer and subjected to further analysis.

## Flow cytometry

Cells collected from peritoneum and spleen, after RBC lysis and Fc receptor blockade with CD16/32 antibody (clone 2.4G2; eBioscience, San Diego, CA, USA) for 15–20 min, were stained with CD11c-PE-cyanine-7 antibody (clone N418; eBioscience) for 30–40 min as per providers recommendation. Following extensive washing, cell samples were run on a LSRFortessa flow cytometer (Beckman Coulter, Pasadena, CA) and data were analyzed using FlowJo software (FlowJo LLC).

## Plaquing assays

Plaquing assays were performed as described (*Fuks et al., 2012*). Briefly, organs were extracted and homogenized on 70 µm cell strainers. The numbers of viable parasites per g of tissue were determined by plaque formation on HFF-1 monolayers.

## Replication assay

mBMDCs were challenged with freshly egressed tachyzoites (ME49/PTG-GFP, 1 h, MOI 2). Cells were washed by centrifugation to remove extracellular parasites and seeded in 96-well plates. Cells were imaged by epifluorescence microscopy to detect number of parasites per vacuoles at 6 h and 24 h post-infection in presence of pharmacological treatments.

## Data mining and statistical analyses

Motility plots were compiled using ImageJ with manual tracking and chemotaxis plugin. X- and y-axes in the plots show distances in µm. For motility assays, box-whisker and scattered dot plots represents median velocities (µm/min) with boxes marking 25th to 75th percentile and whiskers marking 10th and 90th percentiles of the datasets. Gray circles represent velocities from individual cells. Accumulated distances travelled by the cells (% cells tracked are binned at a range of 2 µm distance) are represented as histograms. Bar graphs show mean + SEM. Heat maps represent transcriptional changes in mRNA expression ($2^{-\Delta Ct}$) upon challenge with *T. gondii*. Red and blue color scales indicate percentage increase and decrease in expression, respectively, normalized to expression in unchallenged cells at the same time point, respectively. Data mining and statistical analyses were performed using GraphPad Prism 7.0 (La Jolla, CA, USA). The statistical significance is represented as $p < 0.05$ (*), $p < 0.01$ (**), $p < 0.001$ (***) or non-significant $p \geq 0.05$ (ns).

## Acknowledgements

We thank Prof. Bryndis Birnir (Uppsala University, Sweden), Esther Collantes (Complutense University, Spain) and all members of the Barragan lab for critical discussions. This work was funded by the Swedish Research Council (Vetenskapsrådet, 2018–02411) and the Olle Engkvist Foundation (193-609).

## Additional information

### Funding

| Funder | Grant reference number | Author |
|---|---|---|
| Vetenskapsrådet | 2018-02411 | Antonio Barragan |
| Stiftelsen Olle Engkvist Byggmästare | 193-609 | Amol K Bhandage |

The funders had no role in study design, data collection and interpretation, or the decision to submit the work for publication.

### Author contributions

Amol K Bhandage, Conceptualization, Data curation, Formal analysis, Funding acquisition, Validation, Investigation, Visualization, Methodology, Writing - original draft, Writing - review and editing; Gabriela C Olivera, Sachie Kanatani, Elizabeth Thompson, Investigation, Methodology; Karin Loré, Resources, Supervision; Manuel Varas-Godoy, Resources, Validation, Investigation, Methodology; Antonio Barragan, Conceptualization, Resources, Supervision, Funding acquisition, Writing - original draft, Project administration, Writing - review and editing

### Author ORCIDs

Amol K Bhandage (iD) https://orcid.org/0000-0002-7116-0939
Antonio Barragan (iD) https://orcid.org/0000-0001-7746-9964

### Ethics

Human subjects: The Regional Ethics Committee, Stockholm, Sweden, approved protocols involving human cells. All donors received written and oral information upon donation of blood at the Karolinska University Hospital.

Animal experimentation: All the animal experimentation procedures involving infection and extraction of cells/organs from mice were approved by Regional Animal Research Ethical Board, Stockholm, Sweden in concordance with in EU legislation (permit numbers 9707/2018, 14458/2019 and N 78/16).

### Decision letter and Author response

Decision letter https://doi.org/10.7554/eLife.60528.sa1

Author response https://doi.org/10.7554/eLife.60528.sa2

## Additional files

### Supplementary files

- Supplementary file 1. Primer pair sequences used in real-time quantitative PCR.
- Supplementary file 2. GABA-A R subunits and VDCCs transcribed by phagocytes.
- Supplementary file 3. Sh-RNA construct sequences.
- Transparent reporting form

### Data availability

All data generated or analysed during this study are included in the manuscript and supporting files.

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
