## [Decision Letter]

**Acceptance summary:**

This study demonstrated that GABA regulates the cellular motility of human and murine mononuclear phagocytes. The GABAergic components were present in the phagocytes, such as the metabolic enzymes and transporters, GABA-A receptors and regulators, and voltage-dependent calcium channels. Infection of phagocytes by coccidian parasites activated GABAergic signaling and increased cell motility. Gene silencing and pharmacological modulators of the functional components affected the GABAergic signaling and cell migration. The finding reveals a layer of regulatory role of neurotransmitter machinery in the host-pathogen interplay.

**Decision letter after peer review:**

Thank you for submitting your article "A motogenic GABAergic system of mononuclear phagocytes facilitates dissemination of coccidian parasites" for consideration by *eLife*. Your article has been reviewed by three peer reviewers, one of whom is a member of our Board of Reviewing Editors, and the evaluation has been overseen by Carla Rothlin as the Senior Editor. The reviewers have opted to remain anonymous.

The reviewers have discussed the reviews with one another and the Reviewing Editor has drafted this decision to help you prepare a revised submission.

Summary:

Herein authors demonstrate that GABA regulates the cellular motility of human and murine mononuclear phagocytes. The authors showed that the GABAergic components were present in the phagocytes, such as the metabolic enzymes and transporters, GABA-A receptors and regulators, and voltage-dependent calcium channels. Infection of phagocytes by coccidian parasites activated GABAergic signaling and increased cell motility. Gene silencing and pharmacological modulators of the functional components affected the GABAergic signaling and cell migration. The finding reveals a layer of regulatory role of neurotransmitter machinery in the host-pathogen interplay.

Essential revisions:

1) The study covered the axis from GABA synthesis to signaling, which was comprehensive but also inevitably affected the strength of the evidence on some of the steps. For example, additional and more direct evidence would help consolidate the involvement of the synthesis/production of GADA on the initial and late stages post infection. While the perturbation studies were mainly done with pharmacological modulators or RNAi knockdown, genetic ablation such as using the CRISPR approach in the cell line would strengthen the finding and the conclusion.

2) The authors claim in the Abstract that "This study reveals a hitherto unappreciated role for GABAergic signaling in the host-pathogen interplay between phagocytes and invasive coccidian parasites.". The same group has published two previous papers in PLoS Pathogens, both using murine and human phagocytes, studying this signaling pathway, with sometimes the same assays being performed (measurement of GABA after *T. gondii* infection, pharmacological inhibition of SNAP to block motility etc). The current study completes the picture of the various components of the GABA signaling pathway for parasite-induced motility, but does not discover a novel unappreciated role. The authors have to clearly state and relate all work performed here that is the same or similar to their previous publications. For example, how is the measurement of GABA different from their 2012 publication? Same goes for all inhibitors used in the study that are sometimes the same as in their 2017 paper. What is new here? Is it the details of the subunits found that are novel? Is it using human versus mouse phagocytes for the inhibitor studies? Please clarify in the Results section and in the Discussion.

For the in vivo results (impact of dissemination into brain and organs (spleen etc) but not peritoneum – why is this different to their 2017 publication, where no effect of dissemination into the brain was found (another part of the GABA pathway was inhibited then). This needs to be clarified and discussed.

3) It seems this study is the first time GABA signaling has an effect on *T. gondii* dissemination into the brain. Why was this only analyzed in the acute phase and not the chronic phase?

4) Figure 4, verification of protein levels using western blot or immunostaining and direct determination of GABA downstream signaling such as calcium would help clarify the functional defects of the GABA-A R upon knockdown. Figure 6K and L, western blot to verify the knockdown efficiency is desirable.

5) Figure 1G, a complete characterization of the transcript dynamics from 0-24h during infection would be informative. It seems that the cells of human origin changed more drastically post-infection. Since the cellular motility increased within minutes post-infection, a more detailed detection of GABA in Figure 1H at different time points could also clarify the effect on GABA production post-infection. Also, the regulation on GABA releasing or transportation at the initial phase might be more relevant given the rapid response of cell motility to infection. The expression levels of those components might affect the interpretation of the contribution of GABA on cellular motility at different phases of post-infection.

6) In the proposed model (Figure 10), NKCC1 mediated influx of chloride, which became the substrate of GABA-A R, and regulated GABA signaling. By pharmacological inhibition and gene silencing of NKCC1, the results in Figure 6 suggested that NKCC1 played a role in regulating hypermotility of parasitized phagocytes. However, there were no experiments indicating that NKCC1 functioned through regulating activities of GABA-A R. The authors should establish the connections between NKCC1 and GABA-A R in parasitized phagocytes.

7) To demonstrate that calcium channels were the downstream mediator of GABA signaling, the author showed that perfusion of GABA generated transient cytosolic Ca^2+^ elevations (Figure 8A). The calcium responses in phagocytes infected by coccidian parasites should be measured to support that calcium influx did happen in parasitized phagocytes. In addition, the calcium influx of parasitized phagocytes treated with the GABA-A R blocker should also be shown.

8) SNAP was used as blockade of GABA transporters (Figure 3E). The concentration of GABA in the supernatant should be measured to support that SNAP did block the transport of GABA. The motility and velocities of unchallenged mBMDCs treated with SNAP and other modulators should be measured as negative controls (Figure 3E).

---

## [Author Response]

Essential revisions:1) The study covered the axis from GABA synthesis to signaling, which was comprehensive but also inevitably affected the strength of the evidence on some of the steps. For example, additional and more direct evidence would help consolidate the involvement of the synthesis/production of GADA on the initial and late stages post infection. While the perturbation studies were mainly done with pharmacological modulators or RNAi knockdown, genetic ablation such as using the CRISPR approach in the cell line would strengthen the finding and the conclusion.

An important point related to the secretion of GABA at early and late time-points of infection is brought up. To address this, we have performed kinetic analyses of secreted GABA during 1-24 h. The data show an elevation of GABA in the supernatant shortly after infection and an increase over time, indicating that GABA production starts shortly after parasite invasion and accumulates during infection. The data reinforce the conclusions and is consistent with the observed maintained hypermotility of infected phagocytes over time (Results, subsection “Human and murine mononuclear phagocytes exhibit hypermotility and secrete GABA upon challenge with *T. gondii* and *N. caninum*”, new Figure 1I).

We have further strengthened the evidence for other steps of the GABAergic signaling cascade with new data, reinforcing the conclusions. These are specifically discussed under points 6, 7 and 8 below.

While genetic ablation, for example by CRISPR, in a cell line could strengthen the findings related to specific GABAergic determinants, the main goal of this paper was to describe the general role of GABAergic signaling in primary phagocytes. There is very little or no data on the expression and conservation of the GABAergic system in phagocytes. To this end, it was crucial to use primary cells from various sources and cells from relevant hosts for coccidian infections. Also, the hypermigratory phenotype is weaker or even absent in cell lines which would have complicated the phenotypic analyses and comparisons with primary phagocytes. This likely relates to the finding that, while primary human and murine phagocytes consistently express GABAergic components, we have observed that cell lines of various sources may have altered expression or undetectable expression of specific GABAergic components (our unpublished observations and see also Figure 2—figure supplement 1). On the technical note, we have performed extensive controls in cell lines and primary cells (see also Kanatani et al., 2017). In this study, we target GABAergic signaling in human and mouse phagocytes at 5 different levels in the GABAergic signaling cascade. We use pharmacological inhibitors as guidance for more specific targeting of genes by shRNA. The number of identified targets yielding an abolished migratory phenotype together with reconstitution experiments undoubtedly show that GABAergic signaling drives the hypermigratory phenotype.

2) The authors claim in the Abstract that "This study reveals a hitherto unappreciated role for GABAergic signaling in the host-pathogen interplay between phagocytes and invasive coccidian parasites.". The same group has published two previous papers in PLoS Pathogens, both using murine and human phagocytes, studying this signaling pathway, with sometimes the same assays being performed (measurement of GABA after T. gondii infection, pharmacological inhibition of SNAP to block motility etc). The current study completes the picture of the various components of the GABA signaling pathway for parasite-induced motility, but does not discover a novel unappreciated role. The authors have to clearly state and relate all work performed here that is the same or similar to their previous publications. For example, how is the measurement of GABA different from their 2012 publication? Same goes for all inhibitors used in the study that are sometimes the same as in their 2017 paper. What is new here? Is it the details of the subunits found that are novel? Is it using human versus mouse phagocytes for the inhibitor studies? Please clarify in the Results section and in the Discussion.For the in vivo results (impact of dissemination into brain and organs (spleen etc) but not peritoneum – why is this different to their 2017 publication, where no effect of dissemination into the brain was found (another part of the GABA pathway was inhibited then). This needs to be clarified and discussed.

We agree that this needed additional clarification. GABAergic signaling encompasses a complex signaling cascade with multiple components. Our previous work has described for *T. gondii* infection that (1) mBMDCs and MoDCs secrete GABA, (2) that mBMDCs express some GABAergic components, (3) that pharmacological treatments targeting GABA inhibit hypermotility in mBMDCs and (4) gene silencing of Ca_V_1.3 in BMDCs abolishes hypermotility.

In the current manuscript, we (1) identify and functionally assess the GABA receptor subunits (by selective pharmacological targeting and shRNA) that likely constitute the functional GABA-A receptors, (2) we identify GAD67 the principal GABA synthesis enzyme in phagocytes, (3) we identify NKCC1 as a main regulator of GABAergic signaling in phagocytes. Further, we extend previous findings (which were mostly on murine DCs) to (4) elutriated human myeloid DCs, and monocytes for the first time and (5) extend the concept of GABAergic activation beyond *T. gondii* to *N. caninum*, using 5 separate strains of the two coccidia. We have clarified and highlighted these aspects in the revised discussion.

No single experiment overlaps with our previous publications but sometimes the same pharmacological inhibitors are used together with additional inhibitors and agonists. We have previously identified Ca_V_1.3 as the mediator of calcium fluxes in murine BMDCs (Kanatani et al., 2017). This could have been left out but we reasoned that VDCCs are an important constitutive component of the GABAergic system and therefore, this had to be addressed in a comprehensive approach. However, in the current paper hMoDCs are utilized when studying Ca_V_1.3, which significantly reinforce and extend our previous conclusions in murine cells to human primary cells. There is no experimental overlap.

Our previous work is indicated in the Introduction with references, we have further clarified its reference in the Results section, and bring it up in the Discussion with a summary of the findings in Figure 10. We think this should make it clearer to the reader how the novel findings relate to the two previous papers. We have also provided precision to the above indicated sentence in the Abstract: “The findings reveal a regulatory role for a GABAergic signaling machinery in the host-pathogen interplay between phagocytes and invasive coccidian parasites”.

For the in vivo results (Figure 9), this paper primarily addresses the systemic dissemination of *T. gondii* rather than the passage to the brain. By a novel combination of approaches (competition assay, see our publ. Lambert et al., 2009, and adoptive transfer of shRNA/pharmacologically treated cells), we show that both targeting -here identified- GABA-A R subunits and -here identified- NKCC1 hampers parasite dissemination. The data is in line with our previous data of pharmacological inhibition of GABA synthesis and transportation (Fuks, et al., 2012, not addressed here in vivo). In Kanatani, et al., 2017, pharmacological inhibition of voltage-gated calcium channels was applied (not addressed here in vivo), with an impact on systemic dissemination (blood, MLN, spleen) but non-significant effects on parasite loads in the brain. Thus, different targets and in vivo methodologies are used in each paper. However, the approaches jointly indicate that targeting (1) GABA synthesis/transportation, (2) GABA-A R signaling, (3) GABA-A receptor regulation / NKCC1 or (4) VDCC signaling hamper *T. gondii* systemic dissemination. Approaches (1-3) show an impact on parasite loads in the brain while (4) non-significantly impacts parasite loads in the brain. More important, in the current manuscript, we simultaneously adoptively transferred treated and untreated cells and therefore the untreated cells serve as an internal control for each mouse. These aspects have now been further clarified in the revised Discussion.

3) It seems this study is the first time GABA signaling has an effect on T. gondii dissemination into the brain. Why was this only analyzed in the acute phase and not the chronic phase?

The review raises an interesting question. The review is correct that in a previous paper (targeting VDCC signaling) non-significant differences were observed on parasite loads in the brain (Kanatani et al., 2017). In contrast, an effect on parasite loads in the brain was observed in Fuks et al., 2012. In that paper, we targeted GABA synthesis and transport using pharmacological inhibitors. In the current paper, we target 2 novel functions: GABA-A receptor function and the GABA regulator NKCC1 with an impact on parasite loads in the brain. This raises an important question: Do all GABAergic signaling components impact equally on parasite dissemination? While we believe that different components impact differently or to different extent, we have not tested this extensively. We have clarified this aspect in the revised Discussion, providing references and avoiding excessive speculation.

Regarding testing the role of GABAergic signaling in the model of chronic phase of infection, we have not tested this (yet) for various reasons. The total parasite load in the chronic phase is likely determined by several host and parasite factors, of which the initial invasion is one. The focus here is on the role of GABAergic signaling on the initial systemic dissemination of acute infection. Given the rapid lytic cycle of *T. gondii*, it is unlikely that the *T. gondii* parasites in the CNS during chronic infection relate directly to the initial phagocytes that mediate the initial systemic dissemination and blood transportation of the parasite. However, we think this important question should be addressed in the future, also in light of our recent findings that brain-resident phagocytes (microglia) respond to *T. gondii* infection with hypermigration via GABAergic signaling (Bhandage et al., 2019). This has to be indeed addressed in a different experimental setup which involves different parasite stages (bradyzoites) and different host cell models. Thus, we agree that assessing brain cysts in the chronic phase of infection could add one additional characterization but it would not change the overall conclusions of the paper. Because an effect was observed in the acute phase, we opted not to pursue putative effects in the chronic phase, in line with approved animal ethics protocols. We have highlighted these aspects in the revised Discussion.

4) Figure 4, verification of protein levels using western blot or immunostaining and direct determination of GABA downstream signaling such as calcium would help clarify the functional defects of the GABA-A R upon knockdown. Figure 6K and L, western blot to verify the knockdown efficiency is desirable.

Whenever possible, a verification has been performed in the manuscript, for example by quantification of GABA upon silencing of the synthesis enzyme GAD67 (Figure 5I) or transporter inhibition by SNAP (Figure 5F). However, for other targets, specifically GABA-R subunits, quantifications of knockdown at the protein level have proven to be challenging. We have extensively attempted to quantify expression and knockdown by western blotting and immunofluorescence for GABA-A R subunits and NKCC1. While detection of subunits is feasible in highly-expressing neuronal tissue, weaker bands or no polypeptide bands are observed using purified primary phagocytes. Because expression in phagocytes is likely generally lower compared with neurons, reaching sufficient high numbers of transduced primary phagocytes (which would in theory allow quantitative analyses) is difficult. We have searched the literature for reference, but found that even in neuronal models targeting NKCC1 by shRNA, quantifications are not provided at the protein level (Mejia-Gervacio et al., Neural development, 2011), presumably due to the difficulty to quantify NKCC1 by Western blotting (MW 160-200 kDa, with several splice variants). Moreover, the currently available set of commercial antibodies for GABA-A R subunits yielded inconsistent detection and thus reliable quantifications were not possible. Examples of blots are provided in Author response image 1. The added data (new Figure 8H-I-J) also show the effects of pharmacological stimulation and inhibition of GABA-A receptors at the single cell level and that calcium influx is activated and inhibited, respectively, via GABA-A receptors.

**Author response image 1. sa2fig1:** Representative Western blots show detection of weak or undetectable polypeptide bands for GABA-A R subunits (alpha3, alpha5, rho1) in mBMDCs with commercially available antibodies.

Cell sorting can be used to enrich for transduced populations but was not an option in our case. Here, we were ultimately interested in the functional phenotypic analysis (motility) of transduced cells that had been invaded by Toxoplasma. First, we opted to use cell sorting as a way of enriching for transduced primary cells but our experience is that the harsh flow conditions of cell sorting had a dual effect: (1) Increased cell death/lysis with decreased parasite viability and (2) Stressing the cells with an impact on their migratory behavior making it difficult to control for effects by Toxoplasma infection and treatments. We therefore realized that avoiding this stress moment, while making quantifications of knock-down slightly less precise (likely underestimating knockdown), would be a better reflection of functional gene silencing levels in cells that were phenotypically characterized in motility assays. This did not affect functional motility counts as only transduced cells (positive for the reporter GFP^+^, Figure 4—figure supplement 1A-D) were assessed in the motility assays.

We validated the shRNA constructs and approach in cells known to express GABAergic signaling components (neuronal murine NE4C and human SH-SY5Y) shown in Figure 4—figure supplement 1A-E. We have clarified this in the manuscript (Results).

5) Figure 1G, a complete characterization of the transcript dynamics from 0-24h during infection would be informative. It seems that the cells of human origin changed more drastically post-infection. Since the cellular motility increased within minutes post-infection, a more detailed detection of GABA in Figure 1H at different time points could also clarify the effect on GABA production post-infection. Also, the regulation on GABA releasing or transportation at the initial phase might be more relevant given the rapid response of cell motility to infection. The expression levels of those components might affect the interpretation of the contribution of GABA on cellular motility at different phases of post-infection.

We have expanded our analysis to include additional time points for hMoDCs and Monocytes and these are provided in the new Figure 1H (for GABA synthesis and catabolism enzymes) and new Figure 5D (for GABA transporters).

Additionally, more detailed kinetics (1-3-6-16-24 h) of GABA secretion by hMoDCs from three human donors are provided in the new Figure 1I. The data show an increase of GABA in the supernatant shortly after infection and an increase over time, indicating that GABA production starts shortly after parasite invasion and production/secretion is maintained during infection. Jointly, the data reinforce the conclusions and is consistent with the observed maintained hypermotility of infected phagocytes over time (Results, new Figure 1I).

6) In the proposed model (Figure 10), NKCC1 mediated influx of chloride, which became the substrate of GABA-A R, and regulated GABA signaling. By pharmacological inhibition and gene silencing of NKCC1, the results in Figure 6 suggested that NKCC1 played a role in regulating hypermotility of parasitized phagocytes. However, there were no experiments indicating that NKCC1 functioned through regulating activities of GABA-A R. The authors should establish the connections between NKCC1 and GABA-A R in parasitized phagocytes.

The review raises an important question related to the role of NKCC1 in GABA-A R function and hypermotility. While well-characterized in neuronal cells, the expression of NKCC1 has remained uncharacterized in leukocytes and the present data represent the first evidence of expression by phagocytes, to our knowledge.

To address a direct connection between NKCC1 function and GABA-R we have now performed reconstitution experiments in NKCC1-inhibited cells (bumetanide) using GABA R activation by GABA and muscimol. Contrasting with results on cells with abrogated GABA synthesis (SC; shown in Figure 5G), the results show that GABA or the GABA A receptor agonist muscimol do not reconstitute hypermotility in NKCC1-inhibited cells (new Figure 6O). To confirm this, we have performed stimulation experiments with GABA in shNKCC1 and shGAD67-transduced cells. The data show that upon silencing of GABA production (shGAD67), addition of exogenous GABA reconstitutes hypermotility. In contrast, upon silencing of NKCC1, GABA fails to reconstitute hypermotility (new Figure 6P and Q, Figure 6—figure supplement 1).

Jointly, the data indicate that NKCC1 is needed for optimal GABA-A R function mBMDCs and hMoDCs, reinforcing the signaling model (Figure 10) and in line with regulatory functions on GABA signaling attributed to NKCCs/KCCs in neurons (Bortone and Polleux, 2009, Kaila et al., 2014). The data indicates a connection between NKCC1 and GABA-A R function in parasitized phagocytes (Results, new Figure 6O, P, Q) and we now bring up this aspect in the Discussion.

7) To demonstrate that calcium channels were the downstream mediator of GABA signaling, the author showed that perfusion of GABA generated transient cytosolic Ca^2+^ elevations (Figure 8A). The calcium responses in phagocytes infected by coccidian parasites should be measured to support that calcium influx did happen in parasitized phagocytes. In addition, the calcium influx of parasitized phagocytes treated with the GABA-A R blocker should also be shown.

We have added this data. Indeed, parasitized DCs respond with calcium fluxes to GABA-stimulation in a GABA-A R dependent manner. Moreover, perfusion of SCS (that inhibits β-subunit-containing GABA-A Rs) or picrotoxin (broad inhibitor that blocks all known subtypes of GABA-A Rs) yielded decreased or abolished calcium fluxes in parasitized cells, reinforcing our conclusions on GABA-A Rs and VDCCs. The data have been added to Figure 8H, I, J (Results).

8) SNAP was used as blockade of GABA transporters (Figure 3E). The concentration of GABA in the supernatant should be measured to support that SNAP did block the transport of GABA. The motility and velocities of unchallenged mBMDCs treated with SNAP and other modulators should be measured as negative controls (Figure 3E).

We have performed the suggested experiments with SNAP. The data show that in presence of SNAP, at concentrations applied in neuronal cells, the amount of GABA in the supernatant is consistently reduced by ∼ 60% even after 24 h, supporting the notion that SNAP blocks the transport of GABA. This reinforces the conclusions on the role of GATs. It also provides indications on the involved GABA transporters as SNAP has different affinity for different GATs. SNAP has been shown to have highest selectivity for GAT-2 and GAT-3 and given IC_50_ values are 5, 21 and 388 μM for hGAT-3, rGAT-2 and hGAT-1, respectively. Note also that, in mice, GAT2 corresponds to A12/BGT1, GAT3 corresponds to A13/GAT2 and GAT4 corresponds to A11/GAT3, while GAT1 carries the same name as in humans and rats (A1) (Nelson, 1998; Cohen-Kfir, 2005). Thus, a complete inhibition of GABA transportation by SNAP is not expected. The data has been added to the new Figure 5F (Results).

Data including velocities of all negative controls has now been added in the new Figure 3E and F. The data clarify and reinforce the conclusions (Results).